# The Human T-cell Leukemia Virus capsid protein is a potential drug target

Ruijie Yu[1,2], Prabhjeet Phalora[1,2], Nan Li[1,2], Till Böcking [1,2] &
David Anthony Jacques [1,2] ✉

Human T-cell Leukaemia Virus type 1 (HTLV-1) is an untreatable retrovirus that causes lethal malignancies and degenerative inflammatory conditions. Effective treatments have been delayed by substantial gaps in our knowledge of the fundamental virology, especially when compared to the closely related virus, HIV. A recently developed and highly effective anti-HIV strategy is to target the virus with drugs that interfere with capsid integrity and interactions with the host. Importantly, the first in-class anti-capsid drug approved, lenacapavir, can provide long-acting pre-exposure prophylaxis. Such a property would provide a means to prevent the transmission of HTLV-1, but its capsid has not previously been considered as a drug target. Here we describe high-resolution crystal structures of the HTLV-1 capsid protein, define essential lattice interfaces, and identify a distinct ligand-binding pocket. We show that this pocket is essential for virus infectivity, providing a potential target for future anti-capsid drug development.

Human T-cell Leukaemia Virus type 1 (HTLV-1) was discovered in 1979 and was the first retrovirus identified to cause disease in humans[1]. Approximately 10% of HTLV-1 carriers develop an aggressive and fatal malignancy called Adult T-cell Leukaemia (ATL)[2], and/or a progressive neurological inflammatory condition known as HTLV-1-associated myelopathy/tropical spastic paraparesis (HAM/TSP)[3]. The latter presents with debilitating clinical symptoms including muscle stiffness and weakness, bowel and bladder dysfunction and sexual dysfunction[4]. In addition to these incurable conditions, carriers may also present with other related inflammatory diseases, including pulmonary disease[5], bronchiectasis[6], uveitis[7], and dermatitis[8].

HTLV-1 is found across the world, with regions of endemicity on every inhabited continent. HTLV-1 has been classified into 7 subtypes (A to G), each resulting from zoonotic events that occurred over the last 60,000 years[9]. Subtype A, also known as the cosmopolitan strain, can be found around the world, and is especially prevalent in Japan, where it is estimated to have infected 1.1 million people[10]. Subtypes B, D, E, F, and G are largely restricted to Africa; while subtype C, also known as the Australo/Melanesian subtype, is found in Fiji/ Solomon Islands, Papua New Guinea and the indigenous peoples of central Australia. Indeed, these communities have the highest rates of HTLV-1 in the world, with

recent studies suggesting as many as 40% of the adult population are positive for the virus[11]. HTLV-1c is the most divergent subtype, with estimates suggesting it separated from the rest 40,000–60,000 years ago and entered Australia about 9000 years ago[9,12].

The other retroviruses known to cause disease in humans are HIV-1 and HIV-2, which are responsible for the AIDS pandemic. Over the last four decades, global efforts have seen the realisation of 7 drug classes for the successful treatment of HIV/AIDS. By comparison, there are no effective antiviral treatments available for HTLV-1 and prognosis for patients experiencing symptoms remains poor. Certain antivirals originally developed to treat HIV/AIDS can prevent the spread of HTLV-1 in vitro. These include the nucleoside reverse transcription inhibitors (e.g., Tenofovir) and integrase strand transfer inhibitors (e.g., Dolutegravir), indicating that clinical studies should consider their effectiveness as pre-exposure prophylactics[13–15]. Protease inhibitors show some activity[16], but are generally less potent, and act in the late stage of infection, precluding their use as pre-exposure prophylactics. Recently, the first in vivo study employing humanised mice demonstrated the combination of these drugs alongside MCL-1 inhibition as a possible therapeutic strategy[17]. While clinical studies are yet to be performed, these studies highlight both the potential for HTLV-1 treatment as well

[1]Department of Molecular Medicine, School of Biomedical Sciences, University of New South Wales, Sydney, NSW, Australia. [2]EMBL Australia Node in Single Molecule Science, School of Biomedical Sciences, University of New South Wales, Sydney, NSW, Australia. ✉e-mail: d.jacques@unsw.edu.au

as the limited arsenal of antivirals currently available. Even if successful, without additional antiviral targets, the scope of combination therapy is limited, raising the real possibility of antiviral escape.

The most recently developed strategy to combat HIV infection is 'capsid inhibition'. Studies on the HIV capsid have shown that, rather than being an inert 'shell', the capsid orchestrates many of the early stages of viral infection. During the early post-entry stages of the viral life cycle, the capsid protects the viral genetic material from nucleases, such as TREX[18], and innate sensors, such as cGAS[19]; it engages motor proteins to navigate the cytoplasm[20]; it functions as a semi-permeable reaction chamber to facilitate the process of reverse transcription[21]; it mediates nuclear entry[22–24]; and it partitions into membraneless compartments within the nucleus to direct the integration of viral DNA into specific regions of euchromatin[25]. Remarkably, the capsid is formed from a single protein, CA, which assembles into a fullerene cone comprising ~250 CA hexamers, and exactly 12 CA pentamers[26]. The stability of the assembled capsid is finely tuned as it must 'uncoat' to release viral DNA at the appropriate time and location within the nucleus[27]. The multifunctionality and metastability of the capsid mean that CA mutation often comes at a significant fitness cost to the virus. This 'genetic fragility'[28] and the fact that the capsid carries pockets for host engagement have made the HIV capsid an attractive drug target in recent years.

Lenacapavir, the first in-class anti-capsid drug, was approved for use by the European Union (EU)[29] and the U.S. Food and Drug Administration (FDA)[30] in 2022 for heavily treatment-experienced adults with multidrug-resistant HIV infection. Due to its slow-release kinetics from the injection site and long clearance times, lenacapavir needs only to be administered twice yearly. As such, it is being considered for use in pre-exposure prophylaxis and is the first antiviral to ever show zero infections in a phase 3 HIV prevention trial[31,32].

Lenacapavir was designed based on X-ray crystal structures of the HIV CA protein[33]. It binds primarily to a hydrophobic pocket on the surface of the HIV CA N-terminal domain ($CA_{NTD}$) formed between helices 3, 4, and 5. Additional contacts with helices 8 and 9 on the C-terminal domain ($CA_{CTD}$) of a neighbouring CA monomer ensure the drug preferentially binds the assembled capsid lattice (CA monomer $K_D = 2500 \pm 500$ pM; CA hexamer $K_D = 240 \pm 90$ pM[33]). HIV uses this site to engage with multiple host proteins, including Sec24C[34], CPSF6[35], and the phenylalanine-glycine (FG) repeat domains of the nuclear pore complex[22,35]. By binding to this site, lenacapavir not only inhibits host cofactor engagement, it also promotes aberrant capsid assembly and disrupts mature capsid integrity[33,36]. These multiple mechanisms of action explain the high potency of lenacapavir ($EC_{50} = 23$ pM)[33] and serve to highlight the therapeutic value of targeting the retroviral capsid.

The HTLV-1 capsid is almost identical between subtypes, despite over 50,000 years of divergence (Supplementary Fig. 1a). This remarkable degree of conservation suggests that the HTLV-1 capsid is similarly genetically fragile, highly optimised to the human host, and may also represent a viable target for the development of long-acting prophylactics. As a structural protein with no enzymatic activity, it is necessary to understand the structure of the HTLV-1 CA protein, its lattice formations, and interactions in order to accelerate drug development in this space.

Here, we show that high-resolution crystal structures of the mature-length HTLV-1c CA protein define essential lattice interfaces and identify a distinct ligand-binding pocket. This pocket is essential for virus infectivity, providing a potential target for future anti-capsid drug development.

## Results

### Atomic resolution of HTLV-1 $CA_{NTD}$

Across the known *Orthoretrovirinae*, CA is a two-domain protein with HIV being the best characterised. In this study, we have focused on HTLV-1 subtype C, isolate Aus-GM, isolated from a 67-year-old Western Desert Indigenous Australian male in 2013[12] (GenBank JX891478). The HTLV-1c CA shares 97% sequence identity with subtype A and 32% with HIV-1 M-group CA (Supplementary Fig. 1b). In order to determine the optimal boundary between the N-terminal domain (NTD) and the C-terminal domain (CTD), we predicted the full-length HTLV-1 CA structure using AlphaFold2. The prediction gave high-confidence folds for the individual domains, which were not spatially correlated with each other due to an apparently flexible linker (Supplementary Fig. 2a). Based on residue-level AlphaFold confidence scores, pLDDT (predicted local distance difference test), we determined the junction between the domains to occur between residues S127 and A128. A construct expressing CA 1-127, produced well in *E. coli* and was purified to homogeneity in the absence of an affinity tag by a sequence of ammonium sulfate precipitation, anion exchange, and size exclusion chromatography (Supplementary Fig. 2b). The protein crystallised in space group P1, diffracted to 0.87 Å resolution (Supplementary Table 1), and was solved by molecular replacement using the AlphaFold2 prediction as the search model. The atomic-resolution electron density map unambiguously revealed an N-terminal β-hairpin (residues 1-13) followed by 6 α-helices and an additional $3_{10}$ helix located between helices 4 and 5 (Fig. 1a). Structure alignment with the HIV-1 $CA_{NTD}$ (RMSD = 2.543 Å, Fig. 1b) showed three distinct differences.

Firstly, HTLV-1 CA does not possess α-helix 6, which, in HIV, sits on the exterior surface of the capsid between the β-hairpin and the cyclophilin-binding loop (also known as the 4-5 loop, as it extends off the capsid surface between helices 4 and 5; Fig. 1b). In order to keep the nomenclature consistent between these structures, we have omitted helix 6 from our numbering (Supplementary Fig. 1).

Secondly, HTLV-1 CA is missing the extended 17-residue cyclophilin-binding loop, and instead has a shorter 11-residue sequence that adopts a more compact structure, including a short $3_{10}$ helix. While a cursory inspection of the primary sequence alignment may suggest that residues 92-PLAGP-96 of HTLV-1 are similar to 90-PIAPG-94 of HIV-1, their structures do not align (Fig. 1b and Supplementary Fig. 1). Furthermore, the cyclophilin-binding loop of HIV CA has only been resolved when bound to either Cyclophilin A or Nup358[37] as this loop is conformationally dynamic and contains the isomeric peptide bond (between G89-P90) that can adopt either *cis* or *trans* conformation. Conversely, the 4-5 loop of HTLV-1 CA is resolved in the absence of any bound cofactor and, while it contains two proline residues, all peptides are observed in the *trans* conformation.

Lastly, when HIV virions mature, the proteolytic release of CA from the Gag polyprotein results in an N-terminal proline residue, which forms a salt bridge with a conserved aspartate sidechain (D51). The formation of this bond drives the folding of the N-terminal β-hairpin, which is a hallmark of CA maturity[38]. In some lentiviruses (including HIV-1 M-group, and $SIV_{cpz}$), the β-hairpin has been shown to be able to switch between two states depending on crystallisation pH. In the open state (pH <7) H12 also forms a salt bridge with D51 (Fig. 1c), while in the closed state (pH >7) a water molecule is tetrahedrally coordinated between residues H12, T48, Q50, and D51[21] (Fig. 1d). The consequence of these states is that the β-hairpin 'opens' or 'closes' like an iris above the six-fold axis of the CA hexamer. However, the residue at position 50 varies between lentiviruses and is frequently a tyrosine. Those viruses bearing Y50 (such as HIV-1 O-group, HIV-2 and $SIV_{mac}$) are unable to bind the water molecule, cannot adopt the closed conformation, and consequently have a fixed 'open' conformation of the β-hairpin[19].

HTLV-1 CA forms a structurally equivalent salt bridge between P1-D54 and a similar N-terminal β-hairpin (Fig. 1e). Residues P1 and H12 are well conserved, and T51, K53 and D54 are structurally equivalent to T48, Q50 and D51 of HIV-1 CA. In the HTLV structure, H12 contacts D54, consistent with the structure adopting the 'open' conformation (Fig. 1e). The presence of K53 suggests that HTLV-1 CA is not able to

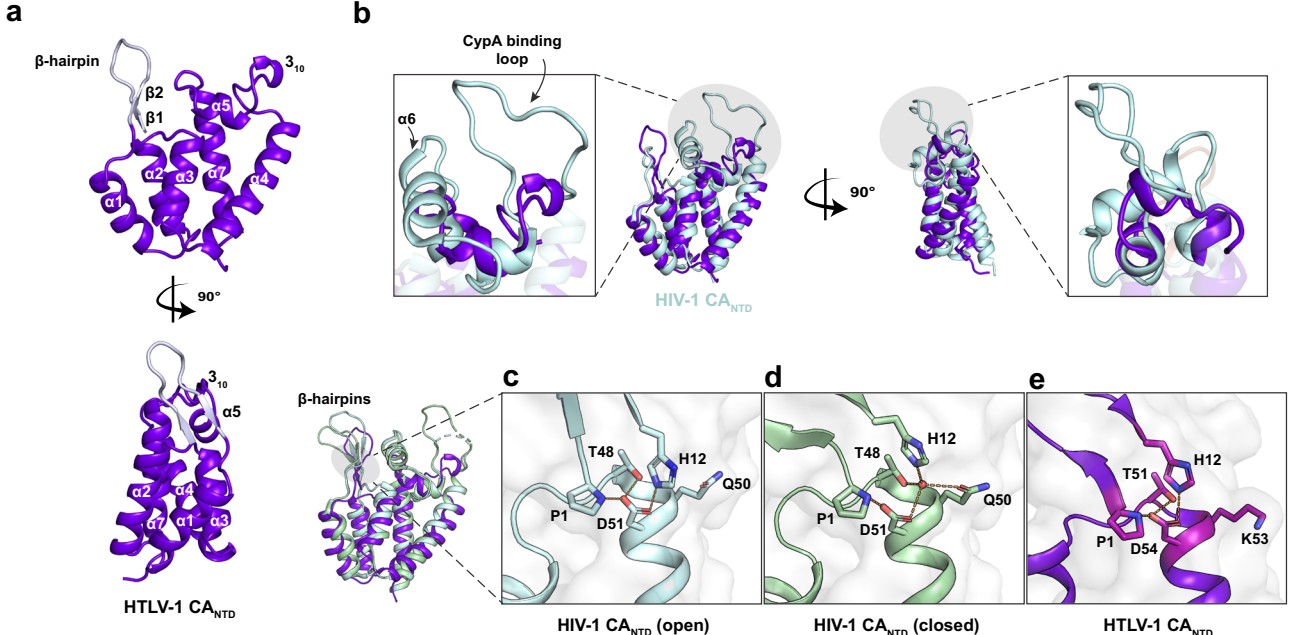

**Fig. 1 | Structural comparison of HTLV-1 CA$_{NTD}$ and HIV-1 CA$_{NTD}$. a** The HTLV-1 CA$_{NTD1-127}$ is labelled with secondary structure elements, including a β-hairpin (pale purple) formed by β-strands 1 and 2, α-helices 1-7, and an additional 3$_{10}$ helix between α4 and α5, a 90° rotated view shown at the bottom. **b** HTLV-1 CA$_{NTD}$ is superposed onto HIV-1 CA$_{NTD}$ (PDB 4XFX, in pale cyan). In HIV-1, the CypA binding loop and α6 are highlighted by the grey shadow with magnified views shown inset. **c–e** HTLV-1 CA$_{NTD}$ is superposed onto HIV-1 CA$_{NTD}$ with open (PDB 5HGL) and closed (PDB 5HGN) states of β-hairpin, showing the residue interactions in the magnified views (inset).

coordinate the water molecule necessary for adopting the closed conformation. We have solved the structure of the CA$_{NTD}$ from both acidic and basic crystallants (see below and methods) and have observed no movement of the β-hairpin. We therefore conclude that, like most retroviruses, the CA β-hairpin is fixed in the 'open' conformation with HTLV-1 unlikely to employ the same dynamic host evasion strategies as reported for the pandemic HIV-1 M-group[19] and its ancestral SIV$_{cpz}$.

### Hexagonal crystal form of HTLV-1 CA$_{NTD}$ reveals capsid lattice packing

In addition to our triclinic crystal form, we also obtained CA$_{NTD}$ crystals in space group P622 that diffracted to a resolution of 2.05 Å (Supplementary Table 1). This crystal form has one molecule in the asymmetric unit with hexagonal 'sheets' generated through crystallographic symmetry (Fig. 2a, b). These sheets have features reminiscent of the HIV CA lattice packing observed in both crystallographic and cryo-EM studies[39,40], wherein the protein makes homotypic interactions to form a hexamer about the 6-fold axis (Fig. 2c). It is notable that the HIV CA$_{NTD}$ is not capable of forming hexagonal lattices on its own, requiring the CTD to provide the necessary inter-hexamer interactions[39,40]. In the case of the HTLV-1 CA$_{NTD}$, inter-hexamer interactions occur at the crystallographic 3-fold axis, and are primarily mediated by residues from helix 4 (Fig. 2d).

Consistent with the mature HIV CA lattice, the HTLV-1 CA$_{NTD}$ has a positively-charged 'pore' or 'channel' at the centre of the hexamer. This feature is formed by six symmetry-related lysine residues at position 18, which are structurally equivalent to the arginine 18 ring found at the centre of the HIV-1 CA hexamer (Fig. 2e, f). In HIV, the R18 pore is only observed in the mature lattice as R18 is pivoted away from the 6-fold channel in the immature lattice. The arginine pore has been shown to bind to polyanions such as inositol hexakisphosphate (IP6) which performs a lattice stabilisation function[41], and deoxynucleoside triphosphates (dNTPs) that are the substrate for encapsidated reverse transcription[21]. It is likely that lysine 18 in HTLV-1 plays similar roles. We

note that the pore is wider than HIV and attempts to co-crystallise with IP6 have not yielded compelling electron density at this site. As such, we cannot rule out the possibility that HTLV-1 binds (an) alternative host factor(s) at this site.

### Full-length mature HTLV-1 CA crystallises as a hexagonal lattice

Previous studies on full-length HTLV-1 CA protein have reported difficulties producing soluble protein, which expressed poorly and degraded rapidly during purification[42]. This was previously resolved by expressing a construct beginning at residue 16, effectively removing the N-terminal β-hairpin, and including an N-terminal His-tag[42]. We found that the full-length HTLV-1c CA protein expressed well in *E. coli*. Initial attempts to purify the protein by ammonium sulfate precipitation followed by cation exchange (at pH 5.0) were successful, but much of the protein precipitated when the pH was raised to 9 (note pI = 8.19). Surprisingly, inspection of the precipitate revealed that, rather than being amorphous aggregate, it comprised small clusters of thin hexagonal crystals. Screening buffer conditions by differential scanning fluorimetry showed a 6-degree increase in melting temperature when the pH was reduced from pH 9 (T$_m$ = 54.6 °C) to pH 6 (T$_m$ = 61.3 °C) (Supplementary Fig. 3a, b). The protein was therefore subsequently purified to homogeneity by size exclusion chromatography and stored in 20 mM HEPES (pH 7.0), 40 mM NaCl.

While readily crystallisable, only one condition yielded diffraction-quality crystals after microseeding. The full-length HTLV-1 CA protein was solved to a resolution of 2.25 Å in space group F222 with three molecules in the asymmetric unit (Supplementary Table 1). Applying a crystallographic 2-fold rotation generated the CA hexamer, packed as sheets, effectively identical to those observed in the hexagonal crystal form of the CA$_{NTD}$ (RMSD = 1.143 Å, Fig. 3a). Despite the high resolution, the electron density for the CTD was of lower quality with a higher B-factor compared with the NTD (Supplementary Fig. 3c), suggesting that it remains flexible relative to the NTD in the crystal. Nevertheless, it was possible to trace the main chain. To obtain higher resolution, we expressed residues A128-L214 (CA$_{CTD}$), which were

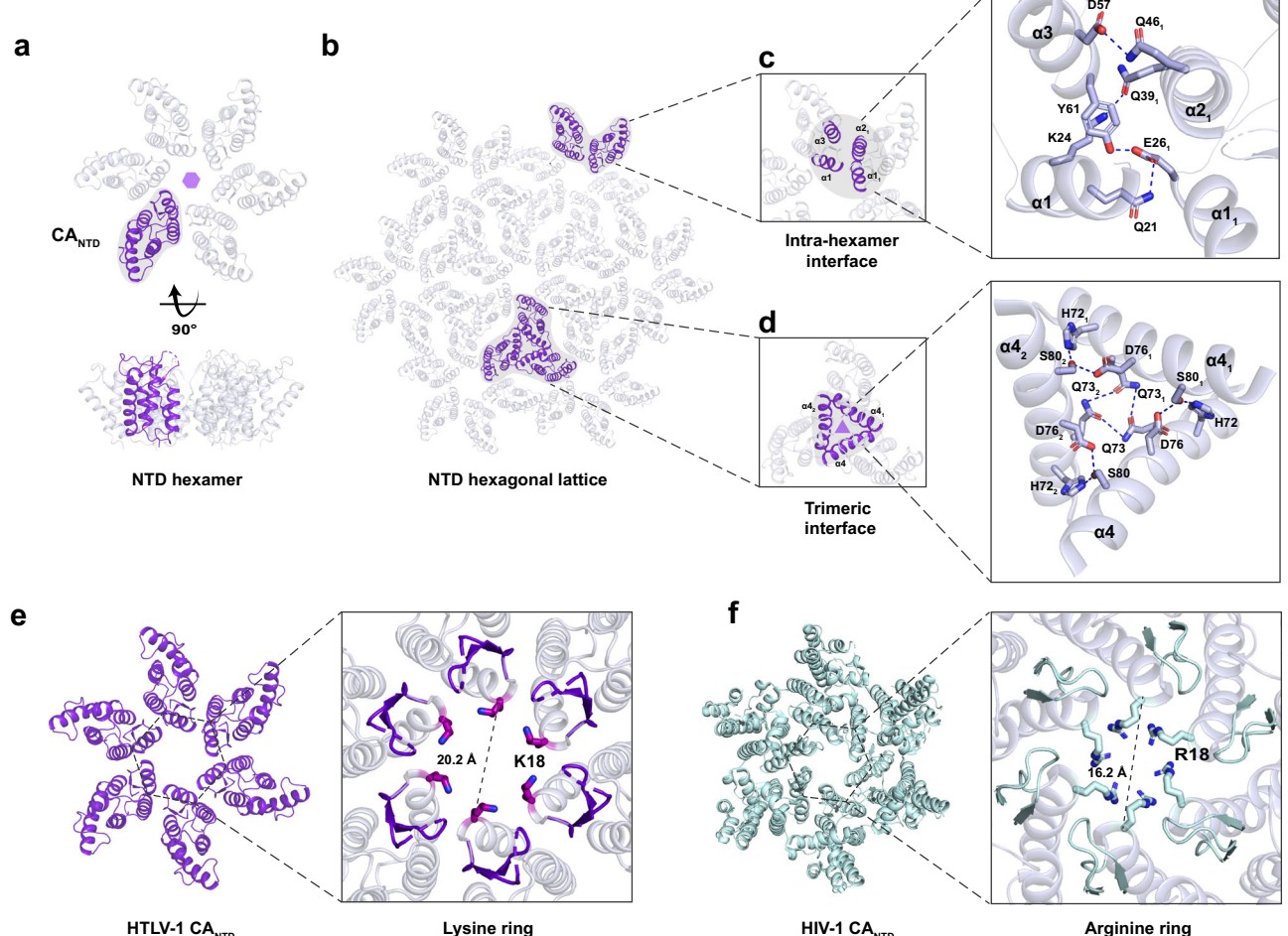

**Fig. 2 | Features of HTLV-1 CA$_{NTD}$ hexagonal lattice and interaction interfaces.**
**a** Within the hexagonal crystal form, the HTLV-1 CA$_{NTD}$ structure generates a hexamer that extends into a hexagonal lattice by crystallographic 6-fold symmetry (**b**).
**c** The intra-hexamer interface reveals the interaction between two NTD subunits, with relevant residues displayed as sticks. **d** The trimeric interface among three inter-hexamers is formed by α-helices 4, with relevant residues shown as sticks.

**e** HTLV-1 CA$_{NTD}$ hexamer contains six lysine residues at position 18, forming a positively charged pore at the centre of the hexamer with a diameter of 20.2 Å.
**f** HIV-1 CA$_{NTD}$ hexamer (PDB 5HGL) has an arginine ring at position 18 with a diameter of 16.2 Å. HTLV-1 CA$_{NTD}$ and HIV-1 CA$_{NTD}$ are shown on the same scale, with the diameter measured from C$_\alpha$ to C$_\alpha$.

purified as a dimer (Supplementary Fig. 3d) and crystallised in space group P2$_1$2$_1$2$_1$ with diffraction to 1.47 Å resolution (Supplementary Table 1). The structure revealed a short 3$_{10}$ helix followed by α-helices 8 to 11 (Fig. 3b, Supplementary Fig. 1). The asymmetric unit comprised 6 copies with two possible dimer configurations: one mediated by helix 9 (Supplementary Fig. 3e), and the other by an interdigitation of residues R188 and H190 in the 10-11 loop (Supplementary Fig. 3f). Comparison to the HIV-1 CA CTD dimer crystal structure suggested the former as the biologically relevant dimer interface, but this was not unambiguous due to structural differences in this region (Supplementary Fig. 3e, g). To improve the CA full-length structure, we resolved it by molecular replacement searching for three independent copies of the refined CA$_{CTD}$ structure as a search model. This unbiased approach returned a CTD dimer virtually identical to that mediated by α-helix 9 in the CA$_{CTD}$ structure (RMSD = 1.580 Å, Fig. 3c), suggesting that this is the biologically relevant dimer interface. This interface is centred on residue Y174. Curiously, while this residue is well-resolved in the electron density map from the CA$_{CTD}$ structure, it makes an aromatic stacking interaction with its counterpart in the dimer partner that necessitates a breaking of the two-fold symmetry (Fig. 3b). The observation that the CTD makes an unusual asymmetric homodimer may partly explain the difficulty in resolving this domain in the full-length structure.

The packing of the HTLV-1 CA into hexagonal 'sheets' is reminiscent of the lattice packing observed in other retroviral capsid crystal structures. In the case of HIV, the hexagonal lattice observed in crystals has been confirmed by cryo-electron tomography with subtomogram averaging[40,43] to closely represent the hexagonal packing of CA within mature capsids. A key difference between the HIV (and ancestral SIV$_{cpz}$) and HTLV lattices is the nature of the inter-hexamer interactions. In the HTLV lattice, the NTD mediates both intra- and inter-hexamer interactions. The CTD is excluded from the NTD plane, but also contributes inter-hexamer interactions through CTD dimerization (Fig. 3d). This results in a closer-packed lattice for HTLV, with a distance between hexamer centres of 74.6 Å, nearly 20 Å shorter than that of the HIV lattice (92.0 Å). In the case of HIV, the CTD is in the same plane as the NTD and is solely responsible for the inter-hexamer interactions (Fig. 3e).

A direct comparison between the HTLV-1c CA full-length crystal structure and those from HIV raises the question of whether our crystal packing more closely represents the mature or immature capsid lattice. On the one hand, the HTLV-1 CA is 'mature' in that it represents the CA sequence that is proteolytically released from the immature gag polyprotein. This cleavage results in an N-terminal proline residue, which is clearly resolved, forming a salt bridge with D54 necessary for β-hairpin formation (Fig. 1e). Furthermore, the arrangement of the

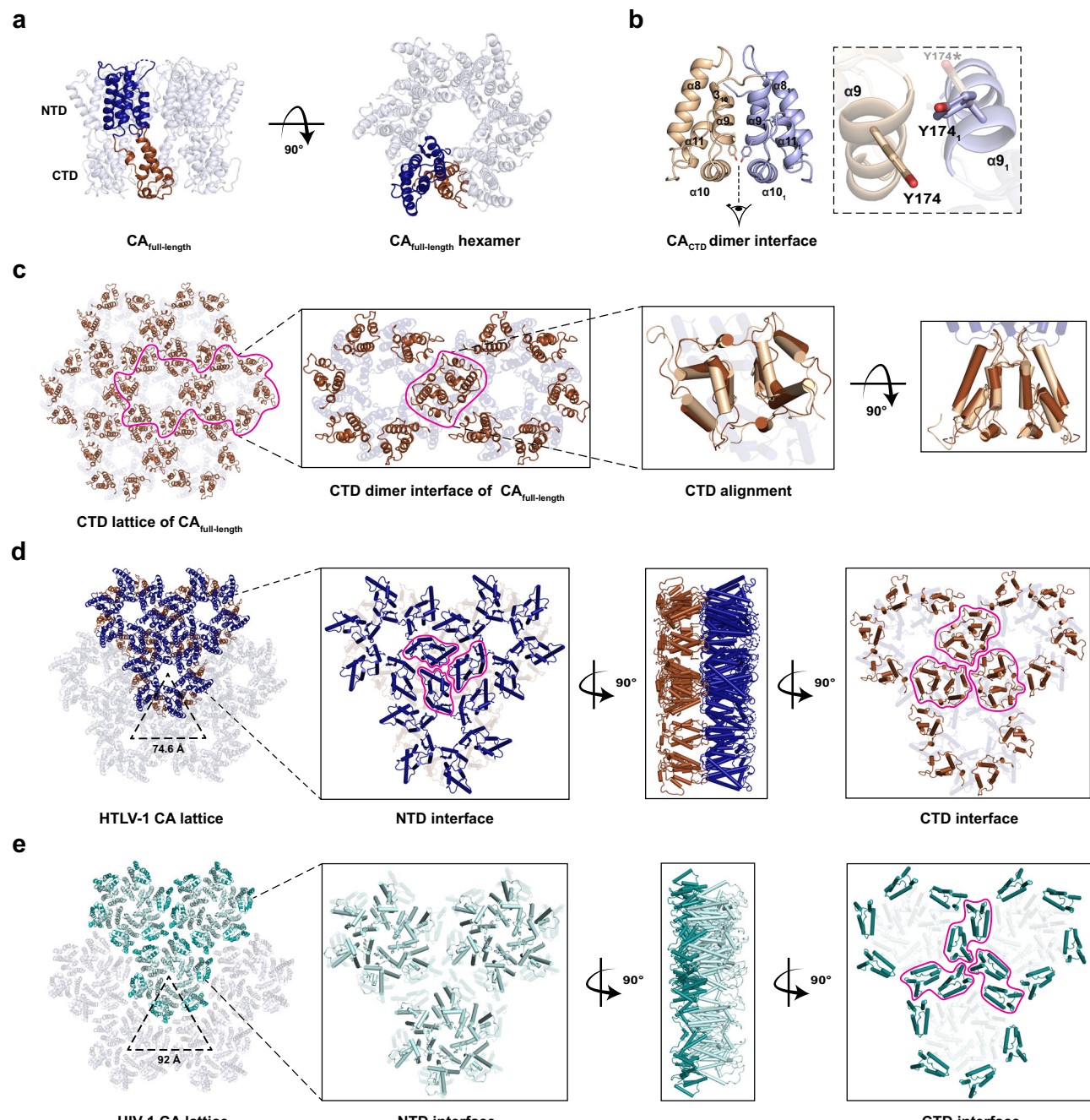

**Fig. 3 | Comparing CA lattice differences between HTLV-1 and HIV-1. a** The crystal structure of full-length HTLV-1 CA comprises domains of NTD (blue) and CTD (brown), shown in side and top views. **b** The crystal structure of $CA_{CTD}$ includes a $3_{10}$ helix followed by α-helices 8-11 and forms a homodimer centred on Y174. A bottom-up magnified view of Y174 shows that π-stacking with the dimer partner residue (Y174_1, light purple) breaks the two-fold symmetry. Y174* denotes the hypothetical symmetric conformation, and is shown for comparison with the actual asymmetric structure. **c** The CA CTD (brown) contributes to CA lattice packing by dimerising with a neighbouring inter-hexamer subunit, which aligns to the crystal structure of CTD dimer (wheat). **d** and **e** The magnified interfaces of HTLV-1 and HIV-1 (PDB 4XFX) (**e**) CA hexagonal lattices are shown as top (NTD face up), side and bottom (CTD face up) views with 90° rotation. The distance between the centres of two hexamers is 74.6 Å in the HTLV-1 CA lattice and 92 Å in the HIV-1 CA lattice.

N-terminal domains positions α-helix 1 at the centre of the hexamer (Fig. 4a). Structures obtained by cryo-electron tomography with subtomogram averaging for authentic HIV particles in both the immature and mature states[40,43], reveals this arrangement occurs only in the mature lattice (Fig. 4c, d). However, the relative position of the HTLV-1c CA NTD and CTD as well as the close lattice packing (Figs. 3d, e, 4a) is not consistent with the HIV mature lattice packing. To resolve this ambiguity, we expanded our comparisons to other retroviruses for

which capsid lattice structures have been derived from cryo-electron tomography with subtomogram averaging. These included Rous sarcoma virus (RSV, an alpharetrovirus)[44,45] and murine leukaemia virus (MLV, a gammaretrovirus)[46] (Fig. 4e–h) as well as a recently published HTLV-1a immature lattice[47] (Fig. 4b). While there is currently no equivalent structure available for the HTLV-1 mature CA lattice, the comparison strongly suggests that our HTLV-1c CA crystal structure packing most closely resembles the immature lattice. In all immature

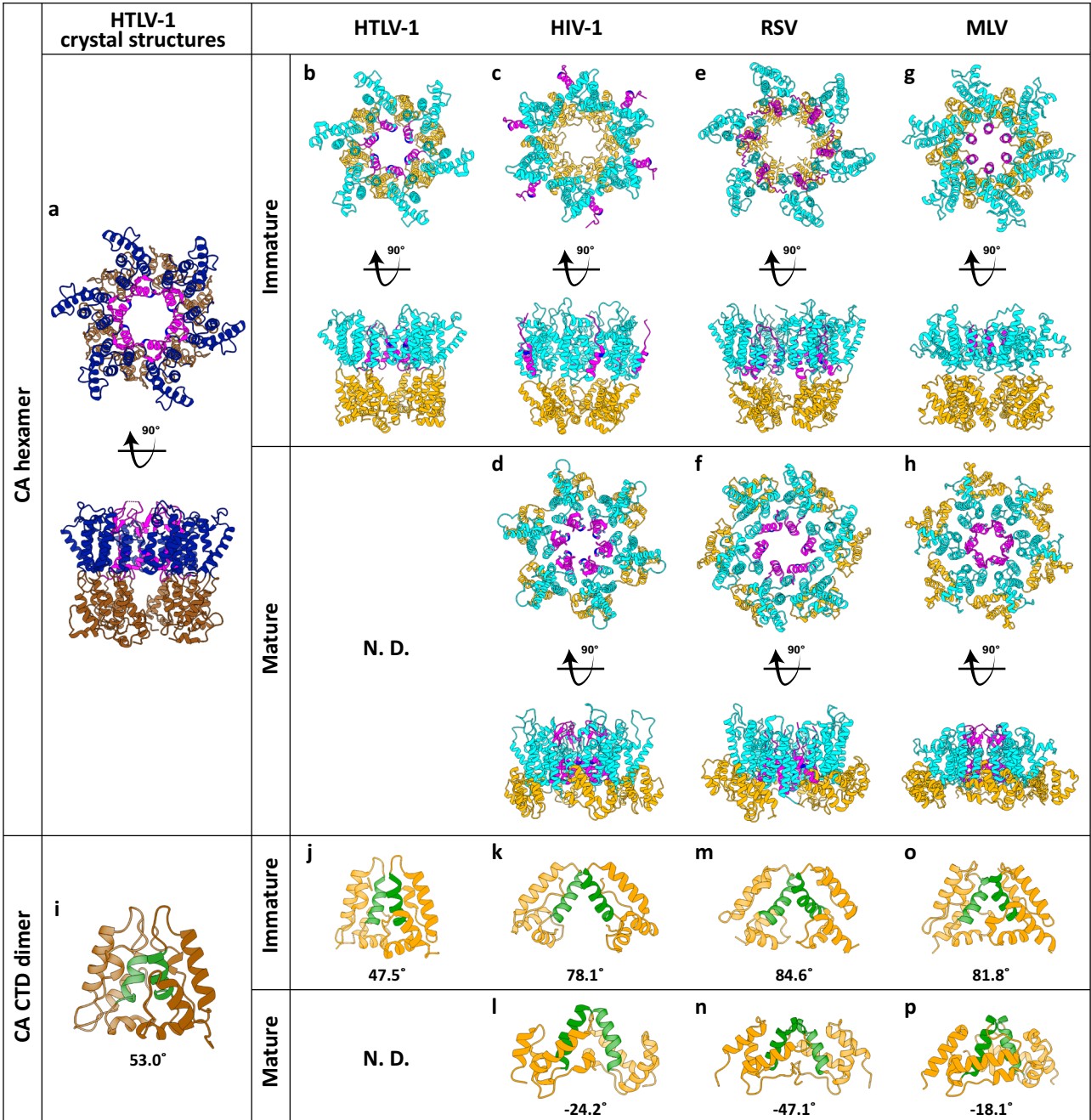

**Fig. 4 | Structural comparison of CA lattice and CA-CTD dimer between HTLV-1 CA crystals and retroviral subtomogram averages.** The crystal structures of the full-length HTLV-1 CA with 6-fold crystallographic symmetry (**a**) and the CA$_{CTD}$ dimer (**i**) are presented in the left column. The subtomogram averages shown for comparison include the immature HTLV-1 CA (**b** and **j**; PDB 8PUG, 8PUH), and immature/mature HIV-1 (**c**, **d**, **k** and **l**; PDB 4USN, 5MCX, 5L93), Rous Sarcoma Virus (**e**, **f**, **m** and **n**; RSV; PDB 5A9E, 7NO2) as well as Murine Leukaemia Virus (**g**, **h**, **o** and

**p**; MLV; PDB 6HWW, 6HWX). In structures derived from cryo-ET with subtomogram averaging, CA$_{NTD}$ is shown in cyan, CA$_{CTD}$ in gold, and the N-terminal β-hairpin connected to α-helix 1 in magenta. The residues of K18 in HTLV-1, R18 in HIV-1, and R21 in RSV are highlighted in navy. **j**–**p** depict the inter-hexamer CTD dimer mediated by helix 9 (green) with the corresponding dihedral angle shown below. Positive dihedral angles denote that the left-hand helix 9 is positioned behind its partner on the right-hand molecule. "N. D." stands for no data.

structures, the CTD sits on a separate plane to the NTD, while in all mature structures, the CTD sits in-plane with the NTD, resulting in an expanded lattice spacing. Importantly, the dramatic rearrangement of the NTD lattice in which α-helix 1 is repositioned toward the central pore upon HIV maturation appears to be a unique property of the lentiviruses and is not observed in RSV, MLV, or HTLV-1.

The inter-hexamer CTD dimer structures were also consistent between our full-length lattice, our isolated CTD crystal structure, and

the recently published HTLV-1a immature lattice[47] (Fig. 4a, i, j). In all cases, the CTD dimer is mediated by helix 9 (Fig. 4j–p, green). In HIV-1, RSV and MLV, the relative arrangement of the two protomers differs between immature and mature lattices. This can be seen most clearly in the dihedral angles defined by the helix 9 pair. Positive dihedral values (in which the left-hand helix 9 sits behind the right-hand helix 9) occur exclusively for the immature packing. Conversely, negative values (in which the left-hand helix 9 sits in front of the right-hand helix 9) occur exclusively in the mature lattice. The positive dihedral value

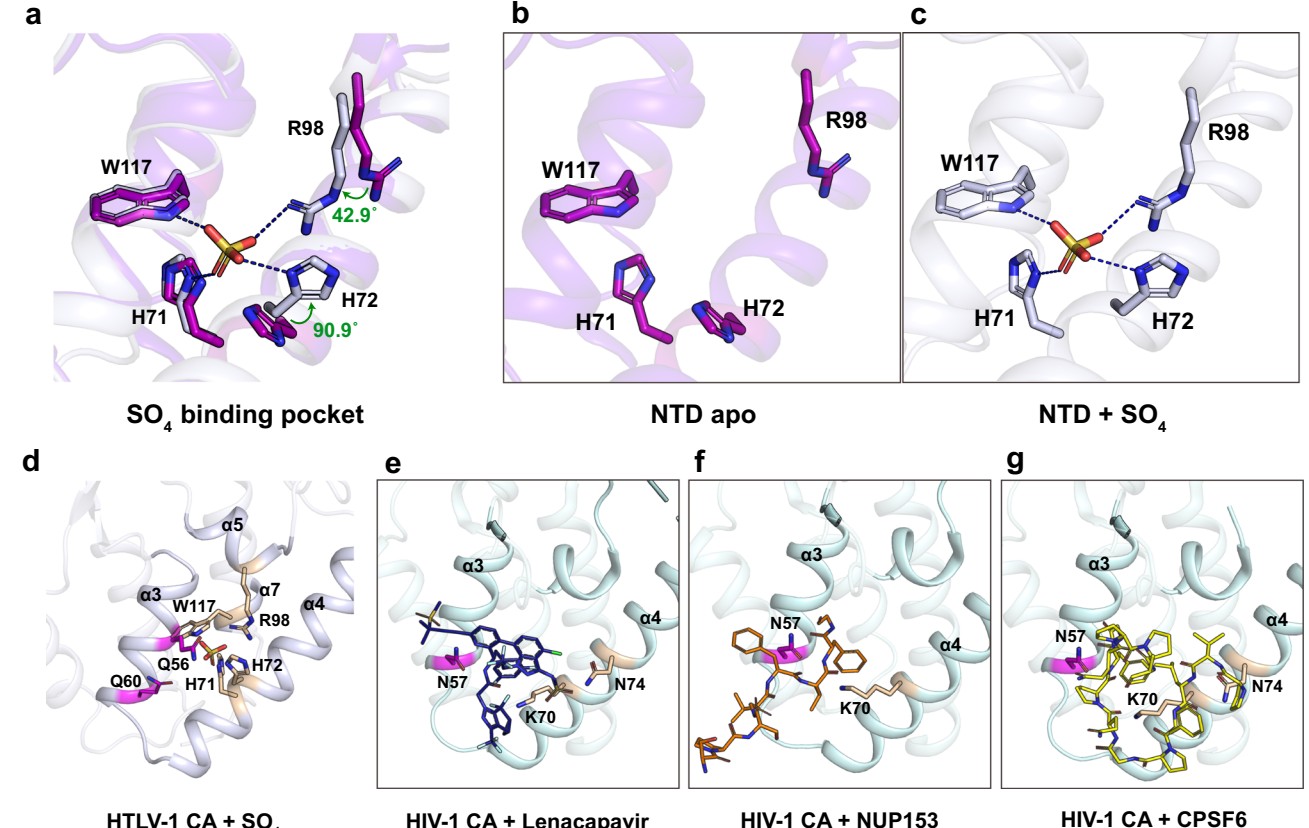

**Fig. 5 | A SO$_4$ binding pocket discovered in HTLV-1 CA$_{NTD}$ orthorhombic crystal form. a** The structural superposition of apo CA$_{NTD}$ (**b**) and SO$_4$-bound CA$_{NTD}$ (**c**) reveals that the SO$_4$ binding site is enclosed by residues H71, H72, R98 and W117. **d** The SO$_4$ binding site is structurally equivalent to the FG-binding site in HIV-1 CA$_{NTD}$. In HIV-1 CA, the FG binding site is occupied by lenacapavir (**e**, dark blue), NUP153 (**f**, orange) or CPSF6 (**g**, yellow), respectively. Key residues interacting with ligands are highlighted in magenta and wheat.

observed for our CTD dimer supports the presence of the immature arrangement in our crystal packing.

Prior studies on HIV have revealed that CA protein rarely interacts with host cofactors outside the context of the capsid lattice, with critical binding sites frequently formed by residues contributed by neighbouring CA monomers. An important advance that proved instrumental in revealing several critical host and drug binding sites[33–35] was the strategic engineering of a disulfide bond enabling isolation of mature HIV hexamers[19,48]. We have attempted a similar disulfide cross-linking strategy for HTLV-1c based on our crystal structures; however, none have resulted in isolatable CA hexamers (Supplementary Fig. 4).

**Crystallant binding pocket may indicate a druggable site**
During screening, a third crystal form of HTLV-1 CA$_{NTD}$ was observed diffracting to 1.47 Å resolution in space group P2$_1$2$_1$2$_1$ with two molecules in the asymmetric unit. In this case, the crystallant included 0.2 M ammonium sulfate. In one of the HTLV-1 CA$_{NTD}$ molecules, a sulfate ion is unambiguously resolved bound to an electropositive pocket on the NTD surface formed by α-helices 3, 4, 5 and 7 (Fig. 5a–c). Upon revisiting the lower-resolution P622 hexagonal crystal form, which also contained ammonium sulfate in the crystallant, a sulfate was observed bound at the same site. The sulfate ion is coordinated by residues H71, H72, R98, and W117 (Fig. 5a, c). The sidechains of Q56, H72 and R98 appear to act as gatekeepers that flip up and down to allow the sulfate entry into the pocket (Supplementary Movie 1).

In the HIV CA$_{NTD}$, the equivalent pocket formed by helices 3, 4, 5 and 7 is hydrophobic and is responsible for recruiting host cofactors

Sec24C, CPSF6, and FG-Nucleoporins[35,49] via their phenylalanine-glycine motifs (Fig. 5e–g). Importantly, this site is the target of the first in class capsid inhibitor, lenacapavir[33], which not only competes for cofactor binding, but also disrupts the capsid ultrastructure[36]. As this site has functional and therapeutic relevance to HIV, we sought to explore whether a biologically relevant compound may bind, and whether our system has the potential for use in ligand screening. The conditions producing the atomic resolution triclinic crystal form were free of sulfate resulting in an apo structure. Using a glutaraldehyde cross-linking strategy, we were able to soak the crystals in 2 M lithium sulfate to obtain a sulfate-bound structure at a resolution of 1.71 Å (Supplementary Fig. 5a), demonstrating proof of concept that this triclinic crystal form is amenable to ligand binding while still maintaining a respectable resolution. As sulfate is unlikely to be relevant to the HTLV infectious process, we performed a similar soak with sodium/potassium phosphate, obtaining a phosphate-bound structure to an almost identical resolution (1.73 Å) (Supplementary Fig. 5b). Interestingly, in crystallants at physiological pH, we observed partial occupancy of both sulfate and phosphate, while full occupancy was achieved under acidic conditions (pH 6 and below). This phenomenon likely reflects the protonation state of the binding pocket, which includes two histidine sidechains (H71 and H72) that typically gain positive charge under acidic conditions.

**Capsid surface residues mediate infectivity and/or particle production**
Our structural data identified several surface residues that may make contacts essential for capsid lattice formation and/or engagement with

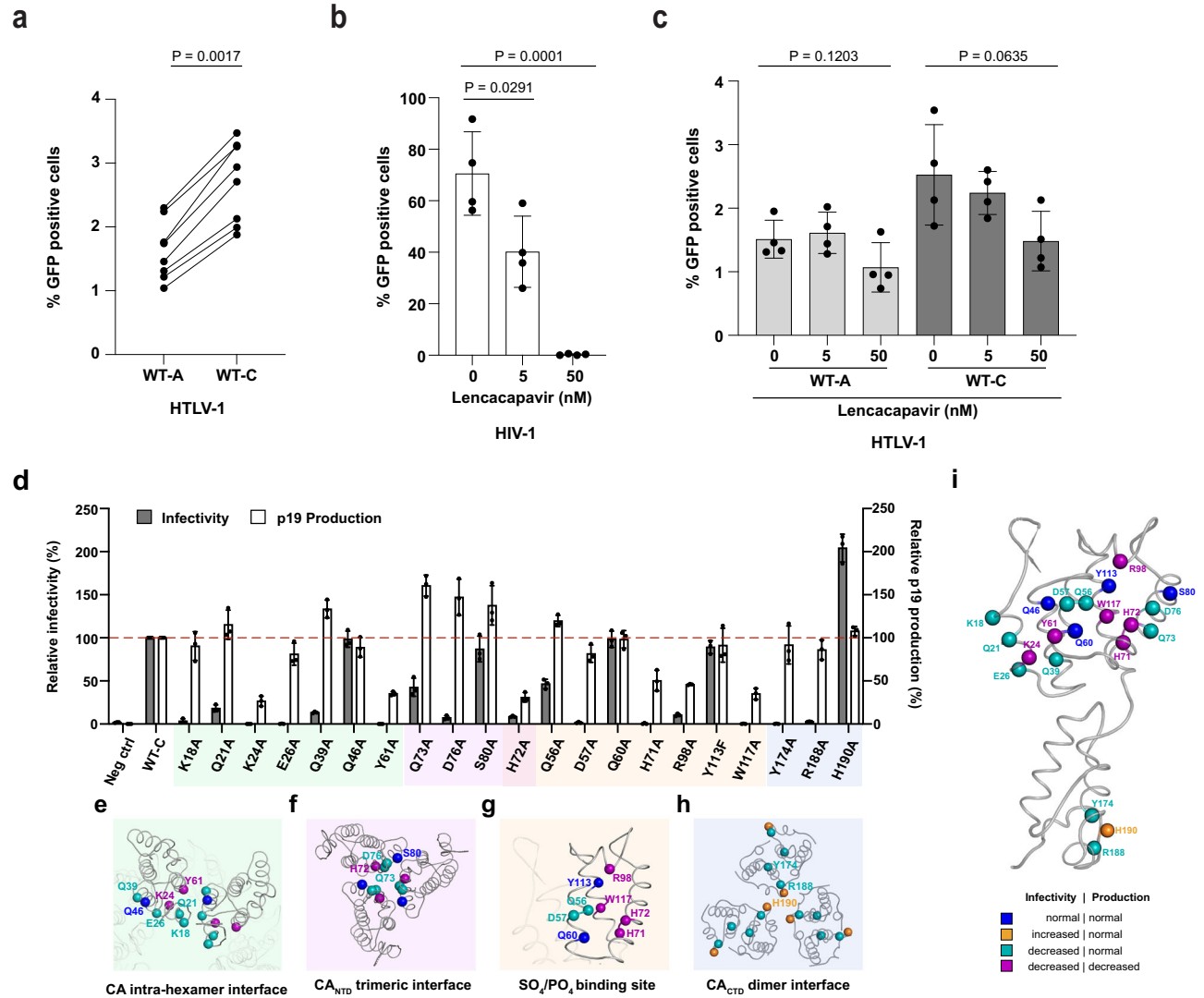

**Fig. 6 | Infection and virus production phenotypes of HTLV-1c CA mutants.**
**a** Graph comparing the infectivity of WT-A and WT-C packaging constructs as measured by GFP reporter expression in a replication dependent infection system. Results are displayed from 8 independent experiments. Statistical analysis was performed using an unpaired two-tailed t-test. Bar graph displaying infectivity of HIV-1 (**b**) and HTLV-1c (**c**) as measured by GFP reporter expression in a replication dependent infection system in the presence of increasing amounts of lencacapavir. Results are displayed as the mean +/− SD from 4 independent experiments. Statistical analysis was performed using an unpaired two-tailed t-test. **d** Bar graph displaying relative infectivity (grey bars) and p19 production (white bars) of 21

HTLV-1c CA mutants, compared to the WT construct, which is set at 100%. Results are displayed as the mean +/- SD from 3 independent experiments. Shaded areas contain residues in the CA intra-hexamer interface (pale green, **e**), CA$_{NTD}$ trimeric interface (pale purple, **f**), SO$_4$/PO$_4$ binding pocket (pale orange, **g**) and CA$_{CTD}$ trimer of dimers interface (pale blue, **h**). **i** HTLV-1c CA mutants, shown in (**d**), mapped onto the structure of the monomeric full-length CA protein, with mutated residues coloured according to infection and virus production phenotypes. Representative gating strategy for the FACS data is provided in Supplementary Fig. 7. Source data are provided as a Source Data file.

external cofactors. Determining the importance of specific residues to the infectious process is complicated by HTLV-1's requirement for cell-to-cell transmission, necessitating co-culture of producer and target cells. To overcome this limitation, we used a previously described replication-dependent infection system based on fluorescent protein reporter expression in HEK-293T cells[50]. Importantly, expression of the reporter is inhibited in producer cells and the fluorescent protein is only produced upon successful reverse transcription and integration in target cells and therefore reports only on cell-to-cell transmission. This single-cycle, plasmid-based system also provides a facile means of screening CA mutants, but was originally developed for the study of HTLV-1 subtype A. For the purposes of this study, we replaced the CA gene with that from subtype C. Cell-to-cell transmission of HTLV-1 is notoriously inefficient, with typical cultures reported to yield 1-3% GFP-positive cells. To our surprise, the inclusion of the HTLV-1 subtype C

CA protein consistently improved infectivity of the reporter virus by approximately 50% relative to the wild-type subtype A (Fig. 6a).

Having established the infection model, we sought to test whether HTLV-1c is sensitive to lencacapavir in this system. While the equivalent HIV cell-to-cell infection was abolished at 50 nM lencacapavir, both HTLV-1 subtypes A and C were completely insensitive to drug treatment (Fig. 6b, c). Our results are consistent with recently published findings using a replication-competent virus, which showed that lencacapavir treatment of either the HTLV-1a producer MT-2 cells or target Jurkat cells showed no discernible antiviral activity[13].

To understand which surfaces on the HTLV-1 capsid might represent vulnerabilities targetable by future anti-capsid compounds, we developed a panel of 21 mutants on the HTLV-1c CA background. These residues were chosen based on their location within one or more of four putative interfaces: the intra-hexamer interface, the NTD

trimeric interface, the CTD trimer of dimers interface, and the sulfate/phosphate binding pocket (Fig. 6e–h). Infectivity was measured by reporter gene (GFP) expression, while particle production was also assessed by p19 ELISA. All mutated residues within the intra-hexamer interface, with the exception of Q46A, substantially reduced viral infectivity. K24A and Y61A concomitantly reduced particle production by more than 50% (Fig. 6d, e), indicating that these two residues are likely indispensable for immature Gag lattice assembly prior to virus budding. Moreover, three mutations along α-helix 4 (H72A, Q73A and D76A), the main interacting interface in the CA$_{NTD}$ trimer, significantly decreased infectivity (Fig. 6d, f), while S80A maintained wild-type levels of infectivity and particle production. Mutants that would remove a direct interaction with the sulfate or phosphate groups (H71A, H72A, R98A and W117A) all effectively abolished infectivity with a concomitant loss of particle production (Fig. 6d, g). Importantly, it should be noted that, in the sulfate-bound state, H72 also spans the CA$_{NTD}$ trimer interface (compare Figs. 2d and 5a). As such, it is conceivable that disruption of the ability of this site to accept a cofactor may have significant consequences for capsid lattice packing through disruption of the trimeric interface. In addition to those sidechains making direct ligand interactions, nearby residues Q56A and D57 in the sulfate/phosphate binding pocket were also found to be important for infectivity, albeit without impacting production, while mutants Q60A and Y113F were indistinguishable from wild type. Y113F was chosen in order to remove only the hydroxyl group of the side chain, which binds to a water molecule, bridging this residue to the sulfate/phosphate groups. Lastly, while we were confident in our assignment of the biologically-relevant CA$_{CTD}$ dimer interface, we decided to test mutants at both this interface (Y174A), and the alternative interface suggested by the interdigitation observed in the CA$_{CTD}$ crystal packing (R188A, and H190A – see Supplementary Fig. 3g). Unsurprisingly, Y174A reduced infectivity to zero, albeit without affecting particle production (Fig. 6d, h). R188A had an identical phenotype. Unexpectedly, H190A resulted in a 2-fold increase in infectivity compared to wild type, with no discernible effect on virus production (Fig. 6a). To the best of our knowledge, this level of infectivity (~7% GFP-positive cells; Supplementary Fig. 6a, b) is the most efficient HTLV-1 single cycle infection result to date.

Overall, we have identified four CA surface mutants phenotypically identical to wild type (Q46A, Q60A, S80A, and Y113F), ten mutants with decreased infectivity despite unchanged particle production (K18A, Q21A, E26A, Q39A, Q56A, D57A, Q73A, D76A, Y174A, R188A), five mutants with decreased particle production and infectivity (K24A, Y61A, H71A, H72A, R98A, W117A), and one mutant that increased infectivity (H190A) (Fig. 6i).

## Discussion

In this study, we have focused on obtaining high-resolution crystal structures of the mature-length HTLV-1c CA protein. Despite previous studies reporting challenges with working with the HTLV-1a CA protein[42], we found that the HTLV-1c CA and its constituent domains are easily expressed, highly stable proteins that routinely produced crystals surpassing the best resolution achieved for the more extensively studied CA from HIV[39,51,52]. We note that some of the residues that differ between subtypes A and C do participate in crystal contacts (such as T119 in the triclinic CA$_{NTD}$ and T146 in the CA$_{CTD}$ crystal forms), which may explain why subtype C is more amenable to crystallisation. Furthermore, in contrast to prior studies, our constructs carry the native N-terminal proline and β-hairpin (residues 1-13). As proline residues carry a positive charge only when found at the N-terminus, P1 can only form the salt bridge with D54 after CA is proteolytically released from Gag. As such, this salt-bridge and formation of the β-hairpin, both of which are well-resolved in our structures, are hallmarks of CA maturity. However, comparison of the hexagonal lattice packing in our full-length CA and CA$_{NTD}$ crystals with matched

pairs of mature/immature CA structures obtained by subtomogram averaging suggests that our full-length structure is more consistent with the immature lattice (Fig. 4). In this regard, the HTLV-1c lattice packing more closely resembles that from the alpha- and betaretrovirus lattices than HIV (lentivirus). While the β-hairpin has clearly formed in our constructs, the lattice packing indicates that proteolytic processing and formation of this structural element is not sufficient to drive mature lattice assembly. As such, it is plausible that the full-length HTLV-1c CA crystal structure represents a maturation intermediate. Future subtomogram averaging studies on the full-length mature capsid will provide further structural insight into the immature-to-mature transition, and possibly also reveal the structure of the CA pentamer, which has remained elusive to date.

As the two retroviruses that cause disease in humans, it is useful to compare and contrast HTLV-1 and HIV. While both CA proteins share the same overall fold, there are significant differences, particularly at the capsid exterior. The fixed position of the HTLV-1c β-hairpin, the absence of helix 6, and the lack of a cyclophilin-binding loop (Fig. 1b) indicate that the cytoplasmic-facing surface of the CA$_{NTD}$ is far less conformationally dynamic than that of HIV. If, like HIV, we assume that the HTLV-1 capsid acts as the host-pathogen interface during the early stages of infection, then it is clear that the two viruses engage with the host in different ways. One aspect where the two virus capsids are known to be functionally different is nuclear entry. Being a lentivirus, HIV is capable of penetrating the nuclear pore complex. It achieves this via specific interaction with FG dipeptide motifs found within the diffusion barrier in the centre of the nuclear pore[22,23]. Furthermore, it is likely disengaged from the nuclear pore complex by a competing interaction with a higher-affinity FG-motif in CPSF6[22,23], which also likely directs the virus to the sites of integration. The hydrophobic FG-binding pocket is formed by helices 3, 4, and 5 on the CA$_{NTD}$, and is also the site targeted by lenacapavir. By contrast, HTLV-1 is a deltaretrovirus and is incapable of entering the nucleus of resting cells, instead gaining access to host chromatin upon dissolution of the nuclear envelope during mitosis. As such, the HTLV-1 capsid does not have a hydrophobic FG-binding pocket and, consequently, it is insensitive to lenacapavir. However, our structures reveal that HTLV-1 does still have a pocket formed between helices 3, 4, and 5, albeit an electropositive one capable of binding to either phosphate or sulfate in vitro. While no capsid-binding cofactors have yet been identified for HTLV-1, it is conceivable that the pocket formed by helices 3, 4, and 5 is a more general 'host recognition' feature that has diversified over the course of retrovirus evolution. In the HTLV-1 case, sulfate is unlikely to be the endogenous cofactor, and inorganic phosphate is too ubiquitous to provide any spatiotemporal regulation to the capsid but could potentially function as a 'pocket factor'[53]. A cryptic phosphopeptide, on the other hand, could conceivably function as a capsid-binding cofactor at this site. In HIV, N57 makes two hydrogen bonds with the main chain peptide of the phenylalanine of the FG-motif (see Fig. 5f, g). In HTLV, the structurally equivalent residue is Q60, which has similar hydrogen-bonding capabilities as the asparagine found in HIV. However, Q60A is phenotypically unchanged from wild type in our single-round infection model. Alternatively, Q56 may function in a similar cofactor-recognition role.

Unlike FG-cofactor or drug binding in HIV, phosphate/sulfate binding in HTLV-1c results in significant structural changes, especially at residues H72 and R98 (Fig. 5 and Supplementary Movie 1). Mutation of the ligand-binding residues abolishes infectivity and reduces particle production (Fig. 6), consistent with a recent 'double-alanine swap' study[47]. One possible explanation would be that a bound cofactor is required for appropriate immature lattice formation prior to budding. It is particularly striking that H72 is so conformationally responsive to sulfate/phosphate binding, while also participating in the trimer interface, and also being essential for infectivity and particle production. As the equivalent site has been exploited for the development of

arguably the most potent and longest-acting anti-HIV drug, our structural and phenotype data suggest that similar attention should be paid to this site in HTLV-1 with a view to the development of capsid-targeting pre-exposure prophylactics. Importantly, each mutant that we have tested is absolutely conserved across known HTLV-1 subtypes (Supplementary Fig. 1). This high degree of conservation suggests that any future capsid-targeting agent would likely be effective across all HTLV subtypes.

Across our panel, we identified ten surface mutants (K18A, Q21A, E26A, Q39A, Q56A, D57A, Q73A, D76A, Y174A, R188A) that reduce infectivity without affecting particle production. These mutants warrant further investigation as this phenotype could result from aberrant capsids and/or loss of an interaction or function important to infection. K18, for example, is the residue responsible for forming the electropositive pore at the six-fold axis, a feature found in all orthoretrovirus genera (with the exception of the gammaretroviruses). This residue likely performs a functional role (e.g., dNTP import) as we see no evidence for IP6 binding in the HTLV-1 case. Likewise, the discovery of a mutant that improves infectivity, H190A, is particularly interesting and also deserves to be the focus of future studies as this is an unusual phenotype to observe during alanine mutagenesis. Given its location in the $CA_{CTD}$, it is not immediately apparent why H190A would improve infectivity. One possible explanation is that H190A improves the fidelity of mature capsid assembly. Given that the residue was identified due to its participation in a homotypic interaction (albeit a crystal contact), it is possible that inappropriate CTD:CTD interactions mediated by H190 occasionally occur during capsid lattice formation. Alternatively, H190A may enable HTLV-1 to escape from a cryptic restriction factor. In either case, the question remains as to why it is advantageous for the virus to maintain a histidine at this position, given that its removal appears to enhance transmission. Nevertheless, the discovery that both HTLV-1c and H190A can improve the single-round infection model raises the possibility that there may be additional ways to improve infectivity in this system (e.g., cofactor expression, restriction factor depletion, alternative mutants). If a single-round infection model could be developed that could achieve higher rates of infection, this would enable high-throughput methods for cofactor identification (e.g., CRISPR screens).

To date, there are no effective antiviral therapies to treat or prevent HTLV-1 infection. While HIV has been the great success story of antiviral development, it took nearly 40 years to realise that the capsid represents one of the best drug targets. The work we present here supports the notion that the same is likely true for HTLV-1: the capsid is vulnerable and has at least one pocket that could be exploited by a small-molecule drug. The robustness of the recombinant CA protein and the ease with which high-resolution crystal structures can be achieved support the feasibility of developing a screening platform amenable to structure-based drug design. Furthermore, understanding the functions and interactions of the capsid during infection will undoubtedly shed light on how to best interfere with the spread of HTLV-1.

## Methods
### Construct design and cloning
HTLV-1, Aus-GM variant of C type, CA full-length protein consists of 214 amino acids with 23.8 kDa in molecular weight. A previous study reported that the CA protein precipitated during purification[42]. To enhance the homogeneity of CA, we designed constructs based on an AlphaFold2 predicted model, showing a flexible hinge as the domain boundary between the NTD and CTD. Consequently, three constructs were generated: (1) full-length protein, HTLV-1 CA, comprising residues of P1-L214; (2) N-terminal domain, HTLV-1 $CA_{NTD}$ (P1-S127), including the intact N-terminal β-hairpin; and (3) C-terminal domain, HTLV-1 $CA_{CTD}$ (A128-L214). The genes were synthesised and subcloned into the pET-30a (+) vector by Genscript, respectively.

### Protein overexpression and purification
The plasmid encoding the full-length HTLV-1 CA, $CA_{NTD}$, or $CA_{CTD}$ was transformed in *E.coli* Rosetta 2 (DE3) pLysS cells, respectively. A single colony was picked and grown in 10 mL LB medium with 30 μg/L Kanamycin and 34 μg/L Chloramphenicol at 37 °C overnight. The starter culture was then used to inoculate 1 L of LB medium at a 1:100 ratio. Cells induced with 1 mM IPTG when $OD_{600}$ reached 0.6 and then incubated at 30 °C for 4 h (180 rpm). Cells were harvested by centrifugation (6000 g, 15 min, 4 °C) and lysed by sonication in precooled lysis buffer, containing 50 mM Tris-HCl (pH 7.0), 50 mM NaCl, 20 mM β-mercaptoethanol, and a tablet of the complete EDTA-free Protease Inhibitor (Roche). The cell debris was removed by centrifugation (29,300 g, 30 min, 4 °C) and the soluble fraction collected for further purification.

The full-length HTLV-1 CA was further purified by ammonium sulfate (20% w/v) precipitation, with the solution stirred 30 min at 4 °C and spun down (29,300 g, 20 min, 4 °C). The pellet was resuspended in 100 mM citric acid (pH 5.0) with 20 mM β-mercaptoethanol and dialysed against 2L of the resuspension buffer for at least three buffer changes. For ion-exchange chromatography, the dialysed protein was briefly centrifuged (29,300 g, 20 min, 4 °C) and the soluble fraction was passed through a 0.45 μM syringe filter (Millipore, Merk) before loading onto the Hiprep 16/10 SP HP column (Cytiva), running with buffer A, 50 mM citric acid (pH 5.0) and buffer B, 50 mM citric acid (pH 5.0) supplemented with 1 M NaCl. The expected eluent was collected and dialysed against 2 L of 20 mM HEPES (pH 7.0) with 40 mM NaCl and briefly centrifuged (29,300 g, 20 min, 4 °C). The soluble fraction was passed through a 0.45 μM syringe filter and loaded onto a HiLoad 26/600 Superdex 200 pg (Cytiva) column for size-exclusion chromatography (SEC), running with the same buffer condition as for dialysis. The expected fractions were concentrated to 12 mg/mL, aliquoted and frozen in liquid nitrogen for storage at −80 °C.

HTLV-1 $CA_{NTD}$ protein was purified as per the CA full-length protein. It was purified onto a HiLoad 26/600 Superdex 75 pg (Cytiva) column. The purified fractions were concentrated to 16 mg/mL, aliquoted, and frozen in liquid nitrogen for storage at −80 °C.

HTLV-1 $CA_{CTD}$ protein was purified through a two-step ammonium sulfate precipitation. Firstly, the junk aggregates were precipitated by 20% w/v ammonium sulfate and removed by centrifugation (29,300 g, 20 min, 4 °C). Next, the target protein retained in the soluble fraction was reprecipitated with 30% w/v ammonium sulfate, pelleted by centrifugation (29,300 g, 20 min, 4 °C), and resuspended in 100 mM citric acid (pH 5.0) with 20 mM β-mercaptoethanol followed by dialysis against 2L of the resuspension buffer at least three buffer changes. The procedure for ion-exchange and SEC followed the protocol of HTLV-1 $CA_{NTD}$. The purified protein was concentrated to 40 mg/mL, aliquoted and frozen in liquid nitrogen for storage at −80 °C.

### Protein crystallisation and X-ray diffraction
All the protein crystals were initially screened by the sitting-drop vapour diffusion method using a crystallisation robot (NT8, Formulatrix) combined with a dispenser for high-throughput preparation of crystal trays (Art Robbins Crystal Phoenix). 100 nL protein was mixed with 100 nL precipitant (commercial kits) on a 96-well tray (MiTeGen), evaporating against 40 μL reservoir precipitant at 20 °C. The screened crystals were manually refined using either sitting- or hanging-drop vapour diffusion method.

The crystal of HTLV-1 CA was optimised by seeding microcrystals into a 4 μL hanging-drop mixture, composed of 2 μL protein solution (5 mg/mL) and 2 μL precipitant (8% v/v 2-propanol, 23% v/w PEG 3350, 0.1 M sodium citrate, pH 6.0), equilibrating against 500 μL of reservoir solution at 20 °C. The optimised crystals were cryoprotected in the mother liquor.

HTLV-1 $CA_{NTD}$ protein was crystallised in three crystal forms with optimisation using sitting-drop vapour diffusion at 20 °C. A 2 μL drop

was prepared with 1 μL protein (10 mg/mL) and 1 μL precipitant and equilibrated against 80 μL reservoir solution. The precipitants prepared for the three crystal forms were: (1) triclinic, 10% ethylene glycol, 10% PEG 8000, 0.1 M HEPES (pH 7.5); (2) orthorhombic, 0.2 M $(NH_4)_2SO_4$, 25% PEG 4000, 0.1 M Na acetate (pH 4.6); and (3) hexagonal, 2 M $(NH_4)_2SO_4$, 2% PEG 400, 0.1 M HEPES (pH 7.5). The cryoprotectant for each crystal form was made of the relative mother liquor with 20% ethylene glycol, 20% MPD, or 10% glycerol, respectively.

Crystals of HTLV-1 $CA_{CTD}$ were also refined using sitting-drop vapour diffusion at 20 °C. A 2 μL drop was prepared with 1 μL protein (20 mg/mL) and 1 μL precipitant, containing 1.5 M $(NH_4)_2SO_4$, 0.1 M $Na_2HPO_4$ (pH 4.2) adjusted with citric acid, and equilibrated against 80 μL of reservoir solution. Crystals were cryoprotected in mineral oil.

All cryoprotected crystals were flash-frozen in liquid nitrogen for X-ray diffraction.

### Cross-linking and soaking of the HTLV-1 $CA_{NTD}$ triclinic crystal

The crystal cross-linking reaction was performed in a 24-well tray with each well containing 0.5 mL crystallising solution and a micro-bridge serving as an isolated reservoir for 2 μL of 25% glutaraldehyde (pH 3.0). A cover slip, hanging a droplet with crystals, was sealed over the top of the well overnight. The cross-linked crystal was briefly washed with mother liquor to stop the cross-linking reaction, then soaked in 2 M ammonium sulfate (pH 6.0) or 2 M sodium/potassium phosphate (pH 6.0) for 4 h. Finally, crystals were cryoprotected in the mother solution with 20% glycerol before flash freezing in liquid nitrogen for X-ray diffraction.

### Data collection, structure solution, and refinement

All datasets were remotely collected and auto-processed at the MX2 beamline (wavelength 0.9537 Å, energy 13 keV) of the Australian Synchrotron[54]. For collecting datasets of the HTLV-1 $CA_{NTD}$ triclinic crystal with atomic resolution, the wavelength was adjusted to 0.7999 Å to obtain a higher energy up to 15.5 keV. The structure of HTLV-1 $CA_{NTD}$ or $CA_{CTD}$ was determined using the AlphaFold2 predicted model as a template for molecular replacement through Phaser and followed by subsequent rounds of refinement in Phenix (version 1.20.1-4487) along with Coot (version 0.9.8.3). The full-length HTLV-1 CA structure was solved by molecular replacement using the solved $CA_{NTD}$ and $CA_{CTD}$ as search models. The quality of the determined structure was evaluated using MOLPROBITY, based on Ramachandran outliers, stereochemical deviations, rotamer outliers, and steric clashes. Structural figures were rendered in PyMOL Molecular Graphics System (version of 2.5.3 Schrödinger) and UCSF ChimeraX (version 1.4). The final statistics for all data collections and refinements are summarised in Supplementary Table 1.

### Protein thermal stability analysis of the HTLV-1 CA

Measurements were performed using a nanoDSF (differential scanning fluorimetry) instrument (Prometheus NT.48), which detects subtle changes in the fluorescence (330/350 nm) of the tryptophan of proteins, offering sensitive label-free monitoring for protein unfolding and refolding processes. This approach allows a rapid screening for optimal protein purification and storage conditions.

The samples were prepared at a final concentration of 3 mg/mL HTLV-1 CA protein in pH 6 (50 mM MES), 7 (50 mM HEPES), 8 (50 mM TRIS) and 9 (50 mM TRIS), respectively, up to a total volume of 20 μL. By capillarity action, each sample was aspirated into standard-treated glass capillaries (NanoTemper) that were loaded into a 48-capillary sample holder. The excitation wavelength was set at 280 nm (20 nm bandwidth), and the LED power was adjusted to produce emission intensities between 330 and 380 nm, ranging from 6000 to 25,000 A.U. The intrinsic fluorescence was measured for 1 to 5 s in the presence of a heat gradient, setting the temperature variance from 20 to 90 °C.

### HTLV-1 CA mutational analysis – infection and viral phenotype

The C-type CA sequence was cloned into pCMVHTLV_delta env (kind gift from Dimitry Mazurov) by Gibson Assembly (NEB) and all subsequent point mutations were introduced by standard overlapping PCR. The sequences of the primers used to generate the mutations are provided in Supplementary Table 2. The DNA sequences of all constructs were verified by whole plasmid sequencing (Primordium Inc).

HEK-293T (ATCC) cells were maintained in Dulbecco's Modified Eagle Medium (DMEM) with high glutamine and glucose, supplemented with 10% FCS. Cells were passaged every 2 days and maintained at 37 °C with 5% $CO_2$. Cells were routinely tested for mycoplasma contamination.

HEK-293T cells were plated at $2.5 \times 10^5$ cells/well in 12-well plates the day before transfection. Transfections were performed using pCMVHTLV-1_delta env, pCRU5-inGFPt (Addgene #60236)) and pVSV-G at a 1:1:0.5 ratio using Fugene HD. For HIV infections, the following plasmids were used for transfections: psPAX2, pUCHR-inGFPt (Addgene #60237) and pVSV-G. Media was replaced 16 hours post-transfection. For lenacapavir (MedChemExpress) experiments, indicated concentrations were added 30 mins prior to transfection and again during media replacement. 72 hours post-transfection, cells and supernatants were harvested for analysis of infectivity and virus production, respectively. For infectivity measurements, cells were stained with Live/Dead Fixable Near-IR Dead Cell Stain (ThermoFisher) for 20 mins at RT and then fixed in 2% paraformaldehyde (Electron Microscopy Sciences) for 20 mins at RT. Samples were run on a BD LSR Fortessa X-20 and the number of GFP-positive cells was enumerated using FlowJo V10 (see Supplementary Fig. 7 for gating strategy).

Virus production was measured using a p19 antigen ELISA kit (Zeptometrix) according to the manufacturer's instructions.

### Reporting summary

Further information on research design is available in the Nature Portfolio Reporting Summary linked to this article.

## Data availability

All data generated or analysed in this study are available within the article and the Supplementary Information. All X-ray diffraction raw data are available via Dryad. Atomic coordinates and electron density maps have been deposited in the Protein Data Bank (PDB) under the accession codes 8ERE, 8ERF, 8ERG, 8ERH, 8ERI, 8TMV, and 8TMW. Source data are provided with this paper.

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

## Acknowledgements

We thank C. Dickson for the critical reading of this manuscript. This work was supported by a National Health and Medical Research Council Ideas Grant (GNT2013215; D.A.J., T.B.) and Wellcome Trust Collaborator Award (214344/Z/18/Z; D.A.J., T.B.). D.A.J. was supported by a UNSW Scientia Fellowship. We also acknowledge the Structural Biology Facility in the Mark Wainwright Analytical Centre – UNSW, funded in part by the Australian Research Council Linkage Infrastructure, Equipment and Facilities Grant: ARC LIEF 190100165. This research was undertaken in part using the MX2 beamline at the Australian Synchrotron, part of ANSTO, and made use of the Australian Cancer Research Foundation (ACRF) detector. We acknowledge the Bedegal people of the Eora nation, the traditional custodians of the land upon which this research took place.

## Author contributions

R.Y. was responsible for construct design, recombinant protein production, crystallization, X-ray data collection, and structure solution with the support of N.L. P.P. adapted the single-cycle infection model for the study of subtype C capsid and performed all infection-relevant experiments. T.B. and D.A.J. conceptualized the project. R.Y. and D.A.J. wrote the manuscript with input from all authors.

## Competing interests

The authors declare no competing interests.
