## [Peer Review File · Nature Communications]

The Human T-cell Leukemia Virus capsid protein is a potential drug target

Corresponding Author: Dr David Jacques

Version 0:

Reviewer comments:

Reviewer #1

(Remarks to the Author)

In their manuscript "The Human T-cell Leukemia Virus capsid protein is a potential drug target" Yu and colleagues present high-resolution crystal structures of HTLV-1 CA constructs. These structures allow the authors to characterize the folds of the N-terminal and C-terminal domains of CA, as well as the hexameric interactions between CA domains in lattice-forming crystals. The authors also study the impact of mutations on viral infectivity and particle production, confirming some of their observed structural interfaces. Analysis of a pocket in the HTLV-1 CANTD and its comparison to its counterpart in HIV-1 lead the authors to the conclusion that HTLV-1 CA is a potential drug target.

Overall, this paper describes high-quality structural data with findings that clearly will be of interest to a wider audience, given the relevance of HTLV-1 as human pathogen.

The main strength of the manuscript is the description of the first high-resolution CA structures obtained via x-ray crystallography (as monomer or within a hexameric lattice). The established protocols can act as a platform for future structure-based drug-design experiments. Indeed, the title of the manuscript is based on the observation that the HTLV-1 CANTD harbors a binding site for phosphate or sulfate. The same location within HIV-1 CANTD is occupied by Lenacapavir, a highly potent HIV-1 CA inhibitor.

However, one could argue that HTLV-1 naturally is a potential drug target, given that any viral protein that is structurally tractable and is required for assembly is a drug target. Also, the reported observation that Lenacapavir is not working against HTLV-1 is not surprising, given the highly tailored interactions this drug has with specific residues in HIV-1, which are not conserved between HIV-1 and HTLV-1.

From the title, I was therefore expecting more insights into what kind of inhibitors the authors envision to work (e.g. via suggesting/testing additional compounds). I do not think such additional experiments are required for this manuscript, but could be part of future work. However, a more detailed introduction and discussion on the current progress of drug development against HTLV-1 would be helpful, together with a more sophisticated discussion of what kind of compounds could be designed against the HTLV-1 pocket.

Another caveat is that the authors are not certain if their hexagonal CA assemblies represent an immature, mature or intermediate state.

Still, all their comparisons are made in comparison to mature HIV-1 CA assemblies. This is based on the presence of a beta-hairpin in their HTLV-1 CA structures which they state is a hallmark of maturity. However, at least in HIV-1 the beta-hairpin is not the main determinant of structural maturation (as shown for example in PMID: 30217893, see Figure 3). In this paper a beta-hairpin is observed in an HIV-1 maturation intermediate forming an immature hexameric CA lattice. In the presented HTLV-1 structures, the presence of the beta hairpin could be simply caused by the combination of the crystallization conditions and lack of the MA domain.

Moreover, in their full-length CA crystal both CA domains are positioned on top of each other, a feature exclusively observed in immature orthoretroviral CA lattices. Only upon maturation CANTD and CACTD rearrange to form lateral contacts and lead to a thinner CA layer.

Also, in the shown crystal structures the CANTD forms trimeric inter-hexamer interactions, similar to the ones previously described in immature HTLV-1 Gag virus-like particles (PMID: 39242978). Such inter-hexameric CANTD interactions have not been observed for any other mature retroviral CA arrangement. Finally, the measured hexamer spacing of ~74 Å is more

similar to an immature lattice (~70-80 Å in HIV-1 and other retroviruses) than to a reported mature lattice spacing. While the authors may be correct when saying “It is possible that, in contrast to HIV, the HTLV-1 capsid undergoes a far less extreme structural rearrangement upon maturation” they still need to compare their full-length CA crystal to the immature HTLV-1 structure that has been published. This should be done via an additional Extended Data Figure as it will allow readers to place the obtained crystal structures into context of other HTLV-1 structure data. In the above statement the authors should also consider mentioning Murine Leukemia virus, which has been shown to have similar immature and mature CANTD lattice conformations (PMID: 30478053).

The focus on HIV-1 for comparison to HTLV-1 is understandable from a drug development perspective (with HIV-1 being the only other human pathogen). However, the peculiarities of HTLV-1 with respect to its assembly and overall lifecycle have been well studied, which make a direct HIV-1 comparison difficult. Hence, the structural analysis falls short in considering other retroviruses which might be more appropriate for a direct comparison. For example, the observation of the CTD dimer interface in HTLV-1 would warrant a comparison to other dimer interfaces in other retroviruses and whether they are based on similar residue contacts.

Other comments:

- Line 32-34: “With such a high-percentage of those infected developing cancer HTLV-1 has been labelled the most potent oncovirus, and has even been argued to be the most carcinogenic substance on the planet⁹”. Despite directly referencing a statement from the cited review, this still seems like a too strong argument unless provide with more context. Since it does not really add to the introduction, the sentence should be removed.
- Line 82-87: The claim about the similarity of HTLV-1 CA between subtypes could be supported with more sequences in the table in Extended Data 1.
- Lines -145: At the moment, the discussion of the open and closed conformation of the beta-hairpin in HTLV-1 lacks biological relevance. The authors should conclude this paragraph with a sentence on what it could mean that there is only one conformation of the beta hairpin, or if they at all consider it relevant.
- Line 152-155 “It is notable that the HIV CANTD is not capable of forming hexagonal lattices on its own, requiring the CTD to provide the necessary interhexamer interactions.” This conclusion about HIV-1 requires a reference, for example PMID: 9765481 or any other paper the authors had in mind.
- Line 167: “Previous studies on full-length HTLV-1 CA protein have reported difficulties producing soluble protein, which expressed poorly and degraded rapidly during purification”. This requires citations of the mentioned studies.
- Line 206-213: “A key difference between the HIV (and SIV) and HTLV lattices is the nature of the inter-hexamer interactions. In the HTLV lattice, the NTD mediates both intra- and inter-hexamer interactions. The CTD is excluded from the NTD plane, but also contributes inter-hexamer interactions through CTD dimerization (Fig. 3d). This results in a closer-packed lattice for HTLV, with a distance between hexamer centres of 74.6 Å, nearly 20 Å shorter than that of the HIV lattice (92.0 Å). In the case of HIV, the CTD is in the same plane as the NTD and is solely responsible for the inter-hexamer interactions (Fig. 3e).”

As explained above, the comparison to mature HIV-1 here seems overly simplistic and also not fitting given that the crystal structures appear to be more immature-like. It is also not clear, why SIV is suddenly mentioned here.

- Lines 265-301: Have any of the selected residues been mutated in earlier studies, either in infectivity or particle release assays. If yes, are the results the same?
- Line 371-372: “Across our panel, we identified ten surface mutants that reduce infectivity without affecting particle production.” Since these mutants are not explicitly listed before, it would aid the reader to state them here in the text.
- Line 716: “Previous work reported that the CA protein precipitated during purification (15).” The provided reference seems incorrect for this statement.
- Line 821: Supplementary Methods are referenced, but not provided.
- Figure 2c,d: The floating residues are a difficult to connect to their respective helices. The authors may want to reconsider the display style, especially since panel 2f displays the helices as ribbons.
- In the provided table for the pdb codes, for P1 with PO4, the cryo column reads “glycerol? Ethylene glycol? MPD? Not sure”. Maybe this could be defined better?

Reviewer #2

(Remarks to the Author)

The Human T-cell Leukemia Virus capsid protein is a potential drug target
Ruijie Yu, Prabhjeet Phalora, Nan Li, Till Böcking, David Anthony Jacques

In the present study, the authors solved the crystal structures of multiple HTLV-1c CA constructs, including full-length, NTD, and CTD at high resolution. The authors provide atomic resolution insights into observed interfaces of NTD, CTD, and full-length constructs and compare partly with HIV-1 CA assembly. In addition, they highlight differences compared to HIV-1 CA, including lack of CypA binding loop, β -hairpin differences, and central pore (IP6 and dNTP binding site in HIV-1 CA), focusing on K18. An SO4/PO4 binding site seems irrelevant for function but provides important insights into the gating mechanism (relevance in the infection cycle has yet to be determined) surrounding H72 and R98. The authors performed an alanine scan within the relevant identified interfaces and identified residues critical for infectivity and p19 production.

Lenacapavir was used to determine possible inhibitory effects. While no significant effect was observed for WT-A and WT-C, it seems that a mild concentration-dependent effect was visible for WT-C, but higher concentrations may be used to determine if this effect is real. A novelty and progress presented in this study include the improved HTLV-1 single-cycle infection assay with increased infectivity levels (7% GFP-positive cells).

Overall, this study is well-written, clear, concise, and structured in the choice of experiments to answer critical structural biology questions. The structural biology of this work is solid. However, this work could be significantly extended regarding mechanistic insights and the impact of the identified mutants on the stability and assembly of HTLV-1 CA. In addition, the novelty of this work is limited in light of previous and recently determined HTLV-1 CA/Gag structures, including the monomeric full-length solution (NMR) structure of HTLV-1 subtype C (PDB: 1QRJ, 1G03). A cryo-ET structure of immature HTLV-1 Gag lattice (PDB: 8PU6/7/8/9/A/B/C/D/E/F/G/H) has been published (doi.org/10.1038/s41594-024-01390-8). However, it is important to acknowledge that the high-resolution structures presented here provide new insights. In its current form, the manuscript is not suitable for Nature Communications unless it is significantly extended into mechanistic insights, including stability and assembly (as suggested below).

Suggested changes:

- 1) Line 75: Change Kd to KD
- 2) Line 80: Include a reference for EC50 = 23 pM
- 3) Line 82: "The HTLV-1 capsid is almost identical between subtypes". This statement requires either accurate reference or a sequence and conservation analysis within this study. Beyond HTLV-1c and a (beyond as shown in the extended figure 1)
- 4) Line 589: Extended Data Fig. 2: Improve the language of figure legend—especially 2b.
- 5) Line 162: Is this larger pore diameter related to and a consequence of the fixed open conformation of the β -hairpin?
- 6) Line 168: Have the authors tried incorporating a tag (His6, for example) at the C-terminus? This could avoid/reduce interference with the β -hairpin (N-terminal) and provide a more stable construct. This is at least the case for expressing HIV-1 hexameric and monomeric constructs.
- 7) Line 184: Provide additional B-factors plots visualized on full-length HTLV-1 to support the assumption of higher flexibility in the Extended Figures section.
- 8) Fig. 3: The CTD interface within the dimer seems to provide some additional insights and differences compared to HIV-1. The authors should highlight these differences beyond Extended Fig. 3d/e (with residue interactions) and compare them to HIV-1 CTD, as they provide insights for packaging and potential target sites.
- 9) Line 223: The authors should use Sulfate or Sulphate consistently throughout the manuscript.
- 10) Fig. 5b and c: Lenacapavir not Lencacapavir
- 11) Fig. 5 and Extended Data Fig.5.: The authors performed a thorough mutational analysis (mostly alanine scan) and identified residues clustered within interfaces that show increased/decreased infectivity and p19 production phenotypes. However, a more mechanistic view and explanations of these results would improve the equality and insights of this manuscript and are required for the journal's standard. For example, the authors should select representative mutants within the 4 interfaces (with increased and decreased infectivity and p19 production) and perform nanoDSF of recombinant CA hexamers, CTD, or NTD to understand if stability contributed to infectivity phenotype. In addition, those mutants could be used to understand assembly. Do those mutants increase or decrease HTLV-1 CA assembly? Those experiments are frequently performed for HIV-1 CA (using monomeric CA) but also for HTLV-1 (doi.org/10.1101/2024.04.07.588439 and doi.org/10.1038/s41594-024-01390-8). In addition, it would be valuable to understand the subcellular localization of representative mutants within infected cells (optional).
- 12) Line 376: The author suggests that K18, similar to R18 in HIV-1 CA, has a dNTP binding and translocation role. The authors should include a possible dNTP binding mode in this study and perform binding studies to improve the mechanistic insights, for example, via fluorescence anisotropy, as previously performed by these authors ([doi: 10.1038/nature19098](https://doi.org/10.1038/nature19098)).

Reviewer #3

(Remarks to the Author)

This manuscript from Yu et al. describes the crystallographic structure of the HTLV-1c capsid (CA) under a variety of different conditions. The authors provide a thorough comparative analysis between their newly determined structures and the previously reported structures of the HIV-1 capsid, which is particularly relevant given the ongoing investigation of lenacapavir as a potential prophylactic intervention for people at risk of contracting HIV. Based on the observation that negatively charged mother liquor components could be unambiguously observed at the HTLV-1c-equivalent of the lenacapavir binding pocket, the authors speculate that this pocket may represent a susceptible druggable site. The authors go on to modify a pre-existing fluorescence-based HTLV-1a infectivity assay and show that a number of the residues that were predicted to form essential lattice contacts are also required for efficient viral infectivity. Overall this manuscript is well-written, the results are clearly presented and the conclusions are supported by the data. It is my opinion that this manuscript should be accepted for publication pending the revisions listed below.

Major comments:

1. The crystallographic analysis appears to have been performed very rigorously. However, there are several discrepancies between the values provided in Table S1 and the PDB validation reports for the deposited structures. In particular, there are consistent deviations between the reported I/sig(I) values and the values calculated by Xtriage during deposition. Table S1

should be revised to reflect these updated values before the manuscript can be accepted for publication.

Minor comments:

1. For ease of reading, it would be beneficial to highlight residues that were found to impact infectivity or p19 production in the sequence alignment of Extended Data Fig. 1. It appears that all of the mutations that were identified which had a negative impact on infectivity were conserved between HTLV-1 subtypes a and c. This high degree of sequence conservation at these putative critical residues would seem to bode well for the development of anti-capsid therapeutics and merits some explicit mention in the Discussion section.

2. RMSD values should be provided when discussing the structural comparisons between HTLV-1 CA NTD and HIV CA (Lines 109-113), as well as when comparing the structure of HTLV-1 CA NTD to the same domain in the context of the HTLV-1 CA hexagonal lattice (Lines 183-184).

Version 1:

Reviewer comments:

Reviewer #1

(Remarks to the Author)

The authors have adequately addressed the reviewer's points and questions and I consider the manuscript suitable for publication.

I congratulate the authors to this very nice work.

Reviewer #2

(Remarks to the Author)

RESPONSE TO REVIEWER COMMENTS

We thank all three reviewers for their constructive feedback, which we feel has led to an improved manuscript. Please find below the original reviewer comments in black and our responses in blue.

Reviewer #1 (Remarks to the Author):

In their manuscript “The Human T-cell Leukemia Virus capsid protein is a potential drug target” Yu and colleagues present high-resolution crystal structures of HTLV-1 CA constructs. These structures allow the authors to characterize the folds of the N-terminal and C-terminal domains of CA, as well as the hexameric interactions between CA domains in lattice-forming crystals. The authors also study the impact of mutations on viral infectivity and particle production, confirming some of their observed structural interfaces. Analysis of a pocket in the HTLV-1 CANTD and its comparison to its counterpart in HIV-1 lead the authors to the conclusion that HTLV-1 CA is a potential drug target.

Overall, this paper describes high-quality structural data with findings that clearly will be of interest to a wider audience, given the relevance of HTLV-1 as human pathogen.

The main strength of the manuscript is the description of the first high-resolution CA structures obtained via x-ray crystallography (as monomer or within a hexameric lattice). The established protocols can act as a platform for future structure-based drug-design experiments. Indeed, the title of the manuscript is based on the observation that the HTLV-1 CANTD harbors a binding site for phosphate or sulfate. The same location within HIV-1 CANTD is occupied by Lenacapavir, a highly potent HIV-1 CA inhibitor.

We thank the referee for this accurate account of our study.

However, one could argue that HTLV-1 naturally is a potential drug target, given that any viral protein that is structurally tractable and is required for assembly is a drug target. Also, the reported observation that Lenacapavir is not working against HTLV-1 is not surprising, given the highly tailored interactions this drug has with specific residues in HIV-1, which are not conserved between HIV-1 and HTLV-1.

We agree with the referee that it is not surprising that Lenacapavir is ineffective against HTLV-1. These data were included for completeness and to formally address what we find to be a frequent question we receive as researchers in this field.

1-1 *From the title, I was therefore expecting more insights into what kind of inhibitors the authors envision to work (e.g. via suggesting/testing additional compounds). I do not think such additional experiments are required for this manuscript, but could be part of future work. However, a more detailed introduction and discussion on the current progress of drug development against HTLV-1 would be helpful, together with a more sophisticated discussion of what kind of compounds could be designed against the HTLV-1 pocket.*

We thank the referee for stating that no such additional experiments are required for this manuscript, and this will form future drug screening work in our lab. One of our goals with

this manuscript is to draw attention to the HTLV-1 capsid as a justifiable drug target and motivate future work in this space. Not being medicinal chemists, we are reluctant to presuppose what form novel anti-HTLV capsid compounds might take and anticipate that future drug screening strategies will reveal relevant molecules as well as potentially other targetable sites in addition to the sulfate/phosphate pocket described here.

In terms of the contextualising the current progress of anti-HTLV-1 drug development, we have added the following to the introduction:

“Over the last four decades, global efforts have seen the realisation of 7 drug classes for the successful treatment of HIV/AIDS. By comparison, there are no effective antiviral treatments available for HTLV-1 and prognosis for patients experiencing symptoms remains poor. Certain antivirals originally developed to treat HIV/AIDS can prevent the spread of HTLV-1 in vitro. These include the nucleoside reverse transcription inhibitors (e.g., Tenofovir) and integrase strand transfer inhibitors (e.g., Dolutegravir), indicating that clinical studies should consider their effectiveness as pre-exposure prophylactics¹³⁻¹⁵. Protease inhibitors show some activity¹⁶, but are generally less potent, and act in the late stage of infection, precluding their use as pre-exposure prophylactics. Recently, the first in vivo study employing humanised mice demonstrated the combination of these drugs alongside MCL-1 inhibition as a possible therapeutic strategy¹⁷. While clinical studies are yet to be performed, these studies highlight both the potential for HTLV-1 treatment as well as the limited arsenal of antivirals currently available. Even if successful, without additional antiviral targets, the scope of combination therapy is limited raising the real possibility of antiviral escape.”

1-2 Another caveat is that the authors are not certain if their hexagonal CA assemblies represent an immature, mature or intermediate state.

Still, all their comparisons are made in comparison to mature HIV-1 CA assemblies. This is based on the presence of a beta-hairpin in their HTLV-1 CA structures which they state is a hallmark of maturity. However, at least in HIV-1 the beta-hairpin is not the main determinant of structural maturation (as shown for example in PMID: 30217893, see Figure 3). In this paper a beta-hairpin is observed in an HIV-1 maturation intermediate forming an immature hexameric CA lattice. In the presented HTLV-1 structures, the presence of the beta hairpin could be simply caused by the combination of the crystallization conditions and lack of the MA domain.

We thank the referee for pointing out the intermediate state (combination of HIV-1 CA-NC) of CA maturation. However, our HTLV-1 CA construct solely contains mature-length capsid protein sequence (Proline 1 to Leucine 214). We rephased the sentence in the Discussion ‘As such, this salt-bridge and formation of the β -hairpin, both of which are well-resolved in our structures, are requirements of CA maturity.’ We address the limited structural comparisons in the following responses.

Moreover, in their full-length CA crystal both CA domains are positioned on top of each other, a feature exclusively observed in immature orthoretroviral CA lattices. Only upon maturation CANTD and CACTD rearrange to form lateral contacts and lead to a thinner CA layer.

Also, in the shown crystal structures the CANTD forms trimeric inter-hexamer interactions, similar to the ones previously described in immature HTLV-1 Gag virus-like particles (PMID: 39242978). Such inter-hexameric CANTD interactions have not been observed for any other mature retroviral CA arrangement. Finally, the measured hexamer spacing of ~ 74 Å is more similar to an immature lattice (~ 70 - 80 Å in HIV-1 and other retroviruses) than to a reported mature lattice spacing.

The reviewer raises a very important point and in response, we have re-evaluated our interpretation of the structures. While specific rewrites are given in the following responses, in short, by including additional lattice comparisons beyond that of HIV, the weight of evidence supports the formation of an immature lattice in our full-length CA crystal structure.

1-3 While the authors may be correct when saying “It is possible that, in contrast to HIV, the HTLV-1 capsid undergoes a far less extreme structural rearrangement upon maturation” they still need to compare their full-length CA crystal to the immature HTLV-1 structure that has been published. This should be done via an additional Extended Data Figure as it will allow readers to place the obtained crystal structures into context of other HTLV-1 structure data.

We thank the referee for their suggestion to compare our structure with the immature HTLV-1 CA structure. We feel this suggestion greatly improves the manuscript. We have rewritten the results to include the following along with a new results figure:

“A direct comparison between the HTLV-1c CA full-length crystal structure and those from HIV raises the question of whether our crystal packing more closely represents the mature or immature capsid lattice. On the one hand, the HTLV-1 CA is ‘mature’ in that it represents the CA sequence that is proteolytically released from the immature gag polyprotein. This cleavage results in an N-terminal proline residue, which is clearly resolved forming a salt bridge with D54 necessary for β -hairpin formation (Fig. 1e). Furthermore, the arrangement of the N-terminal domains positions α -helix 1 at the centre of the hexamer (Fig. 4a). Structures obtained by cryo-electron tomography with subtomogram averaging for authentic HIV particles in both the immature and mature states^{40,43}, reveals this arrangement occurs only in the mature lattice (Fig. 4c,d). However, the relative position of the HTLV-1c CA NTD and CTD as well as the close lattice packing (Fig. 3d,e, Fig. 4a) is not consistent with the HIV mature lattice packing. To resolve this ambiguity, we expanded our comparisons to other retroviruses for which capsid lattice structures have been derived from cryo-electron tomography with subtomogram averaging. These included Rous sarcoma virus (RSV, an alpharetrovirus)^{44,45} and murine leukemia virus (MLV, a gammaretrovirus)⁴⁶ (Fig. 4e-h) as well as a recently published HTLV-1a immature lattice⁴⁷ (Fig. 4b). While there is currently no equivalent structure available for the HTLV-1 mature CA lattice, the comparison strongly suggests that our HTLV-1c CA crystal structure packing most closely resembles the immature lattice. In all immature structures, the CTD sits on a separate plane to the NTD, while in all mature structures the CTD sits in-plane with the NTD resulting in an expanded lattice spacing. Importantly, the dramatic rearrangement of the NTD lattice in which α -helix 1 is repositioned toward to the central pore upon HIV maturation appears to be a unique property of the lentiviruses and is not observed in RSV, MLV, or HTLV-1.”

Fig. 4 Structural comparison of CA lattice and CA-CTD dimer between HTLV-1 CA crystals and retroviral subtomogram averages. The crystal structures of the HTLV-1 CA $CA_{full-length}$ with 6-fold crystallographic symmetry (a) and the CA_{CTD} dimer (i) are presented in the left column. The subtomogram averages compared include the immature HTLV-1 CA (b and j; PDB 8PUG, 8PUH), and immature/mature HIV-1 (c, d, k and l; PDB 4USN, 5MCX, 5L93), Rous Sarcoma Virus (e, f, m and n; RSV; PDB 5A9E, 7NO2) as well as Murine Leukemia Virus (g, h, o and p; MLV; PDB 6HWW, 6HWX). In structures derived from cryo-ET with subtomogram averaging, CA_{NTD} is shown in cyan, CA_{CTD} in gold, and the N-terminal β -hairpin connected to α -helix 1 in magenta. The residues of K18 in HTLV-1, R18 in HIV-1, and R21 in RSV are highlighted in navy. j-p depict the inter-hexamer CTD dimer mediated by helix 9 (green) with the corresponding dihedral angle reported. Positive dihedral angles denote that the left-hand helix 9 is positioned behind its partner on the right-hand molecule. "N. D." stands for no data.

1-4 In the above statement the authors should also consider mentioning Murine Leukemia virus, which has been shown to have similar immature and mature CANTD lattice conformations (PMID: 30478053).

We have included MLV in the new comparison figure (see above). In response to this specific point, the immature and mature lattices of MLV CA-NTD (cyan) are quite different. The immature MLV CA-NTD doesn't present an N-terminal β -hairpin that forms in the mature MLV CA structure. (See illustrative figure below.)

1-5 The focus on HIV-1 for comparison to HTLV-1 is understandable from a drug development perspective (with HIV-1 being the only other human pathogen). However, the peculiarities of HTLV-1 with respect to its assembly and overall lifecycle have been well studied, which make a direct HIV-1 comparison difficult. Hence, the structural analysis falls short in considering other retroviruses which might be more appropriate for a direct comparison. For example, the observation of the CTD dimer interface in HTLV-1 would warrant a comparison to other dimer interfaces in other retroviruses and whether they are based on similar residue contacts.

We thank the referee for suggesting the comparison of CTD dimer interface among other retroviruses. We have incorporated those comparisons in the new Figure 4 and added accompanying text (see above response).

Other comments:

1-6 • Line 32-34: "With such a high-percentage of those infected developing cancer HTLV-1 has been labelled the most potent oncovirus, and has even been argued to be the most carcinogenic substance on the planet⁹". Despite directly referencing a statement from the cited review, this still seems like a too strong argument unless provide with more context. Since it does not really add to the introduction, the sentence should be removed.

This sentence has been removed.

1-7 • *Line 82-87: The claim about the similarity of HTLV-1 CA between subtypes could be supported with more sequences in the table in Extended Data 1.*

We compared additional HTLV-1 CA sequences from subtypes b and g, and provided a new figure (Extended Data Figure 1 panel b) showing the CA sequence alignment among subtypes a, b, c, and g with 99.3 % identity. However, the complete genomes of subtypes d, e, and f are currently unavailable, with only long terminal repeat (LTR) sequences reported, therefore, these subtypes could not be included in this alignment. The Extended Data Fig.1 has been updated showing as below.

a

b

Extended Data Fig. 1 Structure-based sequence alignment of HTLV-1c, HTLV-1a and HIV-1 capsid proteins. **a**, The HTLV-1 CA sequence alignment includes subtypes a, b, c, and g. Residues involved in the CA_{CTD} dimer interface are boxed in yellow. **b**, The alignment of HTLV-1c/a CA (in red) and HIV-1 CA (in blue) showing secondary structural elements. Strictly identical residues are highlighted in red background, and the highly conserved residues are shown in blue and boxed in blue. Residues are marked with stars to indicate their impact on viral infectivity and p19 production: golden stars represent increased infectivity with normal

p19 production; cyan stars, decreased infectivity with normal p19 production; magenta stars, increased infectivity with reduced p19 production; and navy stars, no change in either infectivity or p19 production. The alignment figures were generated using ESPript 3⁵⁴.

1-8 • Lines -145: *At the moment, the discussion of the open and closed conformation of the beta-hairpin in HTLV-1 lacks biological relevance. The authors should conclude this paragraph with a sentence on what it could mean that there is only one conformation of the beta hairpin, or if they at all consider it relevant.*

We have modified this paragraph to conclude: *“We have solved the structure of the CA_{NTD} from both acidic and basic crystallants (see below and methods) and have observed no movement of the β -hairpin. We therefore conclude that, like most retroviruses, the CA β -hairpin is fixed in the ‘open’ conformation with HTLV-1 unlikely to employ the same dynamic host evasion strategies as reported for the pandemic HIV-1 M-group.”*

1-9 • Line 152-155 *“It is notable that the HIV CANTD is not capable of forming hexagonal lattices on its own, requiring the CTD to provide the necessary inter-hexamer interactions.” This conclusion about HIV-1 requires a reference, for example PMID: 9765481 or any other paper the authors had in mind.*

We have added the relevant citations (39 and 40).

1-10 • Line 167: *“Previous studies on full-length HTLV-1 CA protein have reported difficulties producing soluble protein, which expressed poorly and degraded rapidly during purification”. This requires citations of the mentioned studies.*

We have added the relevant citation (42).

1-11 • Line 206-213: *“A key difference between the HIV (and SIV) and HTLV lattices is the nature of the inter-hexamer interactions. In the HTLV lattice, the NTD mediates both intra- and inter-hexamer interactions. The CTD is excluded from the NTD plane, but also contributes inter-hexamer interactions through CTD dimerization (Fig. 3d). This results in a closer-packed lattice for HTLV, with a distance between hexamer centres of 74.6 Å, nearly 20 Å shorter than that of the HIV lattice (92.0 Å). In the case of HIV, the CTD is in the same plane as the NTD and is solely responsible for the inter-hexamer interactions (Fig. 3e).”*

As explained above, the comparison to mature HIV-1 here seems overly simplistic and also not fitting given that the crystal structures appear to be more immature-like. It is also not clear, why SIV is suddenly mentioned here.

In light of the above comparison, we agree that this statement in isolation was overly simplistic. We have modified the language to clarify the inclusion of SIV, and follow this section with the above additional text comparing structures beyond the lentivirus crystal structures.

1-12 • Lines 265-301: *Have any of the selected residues been mutated in earlier studies, either in infectivity or particle release assays. If yes, are the results the same?*

To the best of our knowledge the single point mutants that we have tested are unique to our study. In a 2024 study of from the Schur group (Obr *et al*, reference 44), they tested two double mutants that include three of our singly-mutated residues: H72A/Q73A and L97A/R98A. These resulted in significantly reduced viral particle reduction. In our study, Q73A negatively affected the viral infectivity without reducing without affecting particle production, while H72A and R98A significantly decreased both viral infectivity and production. Our studies are consistent with those previous results. We have added a phrase calling attention to this consistency at line 418 in the discussion.

1-13 • *Line 371-372: “Across our panel, we identified ten surface mutants that reduce infectivity without affecting particle production.” Since these mutants are not explicitly listed before, it would aid the reader to state them here in the text.*

These residues were explicitly listed in the final paragraph of Results. (Lines 354-358: *“Overall, we have identified four CA surface mutants phenotypically identical to wild type (Q46A, Q60A, S80A, and Y113F), ten mutants with decreased infectivity despite unchanged particle production (K18A, Q21A, E26A, Q39A, Q56A, D57A, Q73A, D76A, Y174A, R188A), five mutants with decreased particle production and infectivity (K24A, Y61A, H71A, H72A, R98A, W117A), and one mutant that increased infectivity (H190A) (Fig. 5i).”*)

For clarity, we have restated the relevant residues in parentheses in the discussion (Lines 429-430):

“Across our panel, we identified ten surface mutants (K18A, Q21A, E26A, Q39A, Q56A, D57A, Q73A, D76A, Y174A, R188A) that reduce infectivity without affecting particle production.”

1-14 • *Line 716: “Previous work reported that the CA protein precipitated during purification (15).” The provided reference seems incorrect for this statement.*

We corrected the citation that should be reference 42.

1-15 • *Line 821: Supplementary Methods are referenced, but not provided.*

A new Table S2 was provided (Line 807), which lists the primer sequences used to generate the HTLV-1c CA mutants.

1-16 • *Figure 2c,d: The floating residues are a difficult to connect to their respective helices. The authors may want to reconsider the display style, especially since panel 2f displays the helices as ribbons.*

We changed the display style of helices to cartoon in Figure 2c,d, which is consistent with other structural figures.

1-17 • *In the provided table for the pdb codes, for P1 with PO4, the cryo column reads “glycerol? Ethylene glycol? MPD? Not sure”. Maybe this could be defined better?*

The supplementary file for this information has been corrected.

Reviewer #2 (Remarks to the Author):

The Human T-cell Leukemia Virus capsid protein is a potential drug target
Ruijie Yu, Prabhjeet Phalora, Nan Li, Till Böcking, David Anthony Jacques

In the present study, the authors solved the crystal structures of multiple HTLV-1c CA constructs, including full-length, NTD, and CTD at high resolution. The authors provide atomic resolution insights into observed interfaces of NTD, CTD, and full-length constructs and compare partly with HIV-1 CA assembly. In addition, they highlight differences compared to HIV-1 CA, including lack of CypA binding loop, β -hairpin differences, and central pore (IP6 and dNTP binding site in HIV-1 CA), focusing on K18. An SO₄/PO₄ binding site seems irrelevant for function but provides important insights into the gating mechanism (relevance in the infection cycle has yet to be determined) surrounding H72 and R98. The authors performed an alanine scan within the relevant identified interfaces and identified residues critical for infectivity and p19 production.

2-1 *Lenacapavir was used to determine possible inhibitory effects. While no significant effect was observed for WT-A and WT-C, it seems that a mild concentration-dependent effect was visible for WT-C, but higher concentrations may be used to determine if this effect is real. A novelty and progress presented in this study include the improved HTLV-1 single-cycle infection assay with increased infectivity levels (7% GFP-positive cells).*

We thank the reviewer for their account of our study. In responding to the comment that a mild concentration effect is visible, this effect is not statistically significant. Furthermore, a 2024 study by Kalemera *et al.*, which we cite as reference 13, also showed no effect of lenacapavir on HTLV-1a transmission over a range of 140 pM to 100 nM. At higher concentrations, the solubility of the drug becomes limiting. Given the consistency of our observations with these published results, and structural differences between the HIV and HTLV binding pockets, we have not pursued this dose response curve further.

2-2 *Overall, this study is well-written, clear, concise, and structured in the choice of experiments to answer critical structural biology questions. The structural biology of this work is solid. However, this work could be significantly extended regarding mechanistic insights and the impact of the identified mutants on the stability and assembly of HTLV-1 CA. In addition, the novelty of this work is limited in light of previous and recently determined HTLV-1 CA/Gag structures, including the monomeric full-length solution (NMR) structure of HTLV-1 subtype C (PDB: 1QRJ, 1G03). A cryo-ET structure of immature HTLV-1 Gag lattice (PDB: 8PU6/7/8/9/A/B/C/D/E/F/G/H) has been published (doi.org/10.1038/s41594-024-01390-8). However, it is important to acknowledge that the high-resolution structures presented here provide new insights.*

We thank the reviewer for acknowledging the clarity and quality of the work. A detailed response to the stability and assembly concerns is provided below. In responding to the 'recent' NMR structures, these structures were deposited on the protein databank over 25 years ago and have considerable inaccuracies and differences from our crystal structures. Firstly, they are both structures from subtype A (the reviewer's assertion that they are from subtype C is incorrect). PDB 1QRJ was expressed without the important N-terminal β -hairpin

to include an N-terminal His-tag. Such a truncation does not occur in either the mature or immature forms of the CA protein. Our tag-free purification strategy ensures that the protein produced most accurately reflects the mature cleaved form of the CA protein. PDB 1G03 contains a distorted β -hairpin, which is modeled in a conformation incommensurate with lattice formation. With >27% of residues in outlier regions of the Ramachandran plot, this structure is highly unreliable and would be considered of low quality by modern standards (we actually find the AlphaFold2 predicted structure to be much more accurate). The recent publication of the immature HTLV-1 Gag lattice structures is an important development. We have reconsidered some of our interpretations in light of this paper (described above), which was published after our manuscript was submitted. Our data are highly complementary to these structures, providing the higher resolution and structural insights (lattices, ligand pocket, conformational changes) that cannot be derived from other datasets.

Suggested changes:

2-3 1) Line 75: Change K_d to K_D

K_d was changed to K_D .

2-4 2) Line 80: Include a reference for $EC_{50} = 23 \text{ pM}$

We have added the relevant citation (33).

2-5 3) Line 82: “The HTLV-1 capsid is almost identical between subtypes”. This statement requires either accurate reference or a sequence and conservation analysis within this study. Beyond HTLV-1c and a (beyond as shown in the extended figure 1)

We thank the reviewer for their suggestion. We have compared additional HTLV-1 CA sequences from subtypes b and g, and provided a new figure (Extended Data Figure 1 panel b) showing the CA sequence alignment among subtypes a, b, c, and g with 99.3 % identity. However, the complete genomes of subtypes d, e, and f are currently unavailable, with only long terminal repeat (LTR) sequences reported, therefore, these subtypes could not be included in this alignment.

2-6 4) Line 589: Extended Data Fig. 2: Improve the language of figure legend—especially 2b.

The figure legend was improved as below.

*“Extended Data Fig.2 AlphaFold2-predicted model of HTLV-1 CA and SDS-PAGE analysis of CA proteins. **a**, AlphaFold2-predicted model of HTLV-1 CA is colour-coded by confidence, as assessed by the predicted Local Distance Difference Test (pLDDT). High-confidence regions are shown in navy and cyan, while lower-confidence areas appear in yellow to orange. **b**, Purified protein samples of HTLV-1 $CA_{full-length}$, CA_{NTD} , and CA_{CTD} were separated by SDS-PAGE with a molecular mass ladder on the left (in kilodalton, kDa).”*

2-7 5) Line 162: *Is this larger pore diameter related to and a consequence of the fixed open conformation of the β -hairpin?*

No. The bottleneck for the pore is the residue at position 18, which sits below the β -hairpin. The HIV pore diameter (as measured by the distance between C-alpha atoms of R18) does not change significantly as the pore opens and closes: 16.5 Å (closed) vs 16.2Å (open) (see PDBs 5HGN vs 5HGL). For the same measurement in HTLV-1c (this time at residue K18), the diameter is 20.2 Å. (Note that the C-alpha is chosen to measure the pore diameter as it is a well-defined position. Both the K18 and R18 sidechains exist in multiple conformations, and are not well-determined. As such we have not assigned a numerical value to the true pore diameter.)

2-8 6) Line 168: *Have the authors tried incorporating a tag (His6, for example) at the C-terminus? This could avoid/reduce interference with the β -hairpin (N-terminal) and provide a more stable construct. This is at least the case for expressing HIV-1 hexameric and monomeric constructs.*

We have not tried incorporating a His-tag at the C-terminus. Our purification protocol is specifically designed to avoid the need for affinity tags to ensure that the N- and C-termini of the CA protein are native. This is especially important at the N-terminus, as the N-terminal residue must be a proline for the beta-hairpin to form properly. HIV-1 hexameric and (most) monomeric constructs are similarly produced without affinity tags. Incorporation of a C-terminal His-tag will not change the N-terminus nor justifiably change stability.

It is possible our comments at Line 181 were misinterpreted: 'Previous studies on full-length HTLV-1 CA protein have reported difficulties producing soluble protein, which expressed poorly and degraded rapidly during purification³⁷. This was previously resolved by expressing a construct beginning at residue 16, effectively removing the N-terminal β -hairpin, and including an N-terminal His-tag³⁷.' The 'previous studies' refer to prior published research from another lab, in which 'HHHHHSSGHIEGRHM' was added to replace the initial 1-15 residues of HTLV-1 CA. We argue that this was an inappropriate location for a His-tag as it removed a key structural feature of the protein and likely contributed to prior difficulties in protein purification which we have overcome with our tag-less purification strategy.

2-9 7) Line 184: *Provide additional B-factors plots visualized on full-length HTLV-1 to support the assumption of higher flexibility in the Extended Figures section.*

We thank the referee for suggesting a B-factors plot on CA full-length. The plotted figure has been added as the Extended Data Fig.3c.

2-10 8) Fig. 3: *The CTD interface within the dimer seems to provide some additional insights and differences compared to HIV-1. The authors should highlight these differences beyond Extended Fig. 3d/e (with residue interactions) and compare them to HIV-1 CTD, as they provide insights for packaging and potential target sites.*

In response to this comment, and the similar suggestion made by reviewer 1 (1-5 above) we have included the new Figure 4 (with subpanels i-j specifically focussed on the CTD). We have also included the below additional panels to Extended Data Figure 3 to explicitly highlight the differences between the HTLV-1c and HIV CTD dimer interfaces.

2-11 9) Line 223: The authors should use Sulfate or Sulphate consistently throughout the manuscript.

The spelling has been corrected to consistently use 'sulfate'.

2-12 10) Fig. 5b and c: Lenacapavir not Lencacapavir

Corrected.

2-13 11) Fig. 5 and Extended Data Fig.5.: The authors performed a thorough mutational analysis (mostly alanine scan) and identified residues clustered within interfaces that show increased/decreased infectivity and p19 production phenotypes. However, a more mechanistic view and explanations of these results would improve the equality and insights of this manuscript and are required for the journal's standard. For example, the authors should select representative mutants within the 4 interfaces (with increased and decreased infectivity and p19 production) and perform nanoDSF of recombinant CA hexamers, CTD, or NTD to understand if stability contributed to infectivity phenotype. In addition, those mutants could be used to understand assembly. Do those mutants increase or decrease HTLV-1 CA assembly? Those experiments are frequently performed for HIV-1 CA (using monomeric CA) but also for HTLV-1 (doi.org/10.1101/2024.04.07.588439 and doi.org/10.1038/s41594-024-01390-8). In addition, it would be valuable to understand the subcellular localization of representative mutants within infected cells (optional).

We thank the referee for their suggestion. We have attempted to introduce intermolecular disulfide bonds to assemble the monomeric HTLV-1 CA into stabilised hexamers. However, unlike HIV-1 CA, the disulfide cross-linked HTLV-1 CA does not assemble into either hexamers

or pentamers under high ionic strength conditions (see size-exclusion data below and in Extended Data Figure 4). As it is not possible to isolate the necessary reagent to make the measurements (and the papers quoted do not describe such a reagent) we have not attempted to determine the effect of mutation on hexamers. While the quoted paper from the Mansky lab do report observable assemblies for HTLV-1a CA, we have not seen comparable assemblies for HTLV-1c CA. We are currently exploring these differences in *in vitro* assemblies between subtypes (both in terms of structures and kinetics), and mutational studies will form part of this future work.

a

b

Extended Data Fig. 4 Size-exclusion chromatography of cross-linked HTLV-1 CA by cysteine substitution. **a**, Comparisons of SEC elution peaks among the HIV-1 CA hexamer and assembled HTLV-1 CA mutants at pH 7.0 or **(b)** pH 9.0.

2-14 12) Line 376: *The author suggests that K18, similar to R18 in HIV-1 CA, has a dNTP binding and translocation role. The authors should include a possible dNTP binding mode in this study and perform binding studies to improve the mechanistic insights, for example, via fluorescence anisotropy, as previously performed by these authors (doi: 10.1038/nature19098).*

We thank for the referee's suggestion. However, the hexamer CA is mandatory to present the K18 hexameric ring to measure dNTP binding (as we previously described in the quoted Nature paper). As this reagent cannot currently be isolated, these measurements are not possible (see above).

Reviewer #3 (Remarks to the Author):

This manuscript from Yu et al. describes the crystallographic structure of the HTLV-1c capsid (CA) under a variety of different conditions. The authors provide a thorough comparative analysis between their newly determined structures and the previously reported structures of the HIV-1 capsid, which is particularly relevant given the ongoing investigation of lenacapavir as a potential prophylactic intervention for people at risk of contracting HIV. Based on the observation that negatively charged mother liquor components could be unambiguously observed at the HTLV-1c-equivalent of the lenacapavir binding pocket, the authors speculate that this pocket may represent a susceptible druggable site. The authors go on to modify a pre-existing fluorescence-based HTLV-1a infectivity assay and show that a number of the residues that were predicted to form essential lattice contacts are also required for efficient viral infectivity. Overall this manuscript is well-written, the results are clearly presented and the conclusions are supported by the data. It is my opinion that this manuscript should be accepted for publication pending the revisions listed below.

We thank the referee for the positive evaluation of our manuscript and accurately account of our study. We appreciate comments and have addressed all the suggested revisions, as detailed below.

Major comments:

3-1 1. *The crystallographic analysis appears to have been performed very rigorously. However, there are several discrepancies between the values provided in Table S1 and the PDB validation reports for the deposited structures. In particular, there are consistent deviations between the reported $I/\sigma(I)$ values and the values calculated by Xtriage during deposition.*

Table S1 should be revised to reflect these updated values before the manuscript can be accepted for publication.

We appreciate the reviewer checking the PDB validation files. We have investigated each deposition to understand this discrepancy. We have found that those structures for which there is significant deviation between our reported $I/\sigma(I)$ and those in the validation report have occurred when we have uploaded intensity values during the deposition. While not completely transparent, it appears that during validation the PDB converts these intensities into amplitudes and then uses these values to re-estimate the intensities and errors. This unusual process seems to be resulting in erroneous $I/\sigma(I)$ values, but only in datasets for which intensities are submitted. It is also unclear from the PDB validation report what the width is of the high-resolution shell used for calculating these values. We have rechecked all our reported values in AIMLESS, XTRIMAGE and XDS, and are confident in the values we have reported. Unfortunately, the $I/\sigma(I)$ generated in the PDB validation is not a reliable value in certain circumstances. All our structure factor files are publicly available for independent scrutiny.

Minor comments:

3-2 1. *For ease of reading, it would be beneficial to highlight residues that were found to impact infectivity or p19 production in the sequence alignment of Extended Data Fig. 1. It appears that all of the mutations that were identified which had a negative impact on infectivity were conserved between HTLV-1 subtypes a and c. This high degree of sequence conservation at these putative critical residues would seem to bode well for the development of anti-capsid therapeutics and merits some explicit mention in the Discussion section.*

We thank the referee for their suggestion. We have improved the layout of Extended Data Fig. 1, with tested residues denoted by stars coloured according to their mutant phenotype (colour scheme consistent with Fig. 6). We have also included the following text in the discussion: *“Importantly, each mutant that we have tested is absolutely conserved across known HTLV-1 subtypes (Extended Data Fig. 1). This high degree of conservation suggests that any future capsid-targeting agent would likely be effective across all HTLV subtypes.”*

3-3 2. *RMSD values should be provided when discussing the structural comparisons between HTLV-1 CA NTD and HIV CA (Lines 109-113), as well as when comparing the structure of HTLV-1 CA NTD to the same domain in the context of the HTLV-1 CA hexagonal lattice (Lines 183-184).*

We thank the referee for pointing out that RMSD values should be provided. The RMSD values have been calculated using Coot and appropriately reported in the revised manuscript (Lines 119, 198).